# Riemannian TransE:
# Multi-relational Graph Embedding in Non-Euclidean Space

## Abstract

Multi-relational graph embedding which aims at achieving effective representations with reduced low-dimensional parameters, has been widely used in knowledge base completion. Although knowledge base data usually contains tree-like or cyclic structure, none of existing approaches can embed these data into a compatible space that in line with the structure. To overcome this problem, a novel framework, called Riemannian TransE, is proposed in this paper to embed the entities in a Riemannian manifold. Riemannian TransE models each relation as a move to a point and defines specific novel distance dissimilarity for each relation, so that all the relations are naturally embedded in correspondence to the structure of data. Experiments on several knowledge base completion tasks have shown that, based on an appropriate choice of manifold, Riemannian TransE achieves good performance even with a significantly reduced parameters.

## 1 Introduction

### 1.1 Background

Multi-relational graphs, such as social networks and knowledge bases, have a variety of applications, and embedding methods for these graphs are particularly important for these applications. For instance, multi-relational graph embedding has been applied to social network analysis (Krohn-Grimberghe et al., 2012) and knowledge base completion (Bordes et al., 2013). A multi-relational graph consists of entities $\mathcal{V}$, a set $\mathcal{R}$ of relation types, and a collection of real data triples, where each triple $(h, r, t) \in \mathcal{V} \times \mathcal{R} \times \mathcal{V}$ represents some relation $r \in \mathcal{R}$ between a head entity $h \in \mathcal{V}$ and a tail entity $t \in \mathcal{V}$. Embedding a multi-relational graph refers to a map from the entity and the relation set to some space. Mathematical operations in this space enable many tasks, including clustering of entities and completion, prediction, or denoising of triples. Indeed, completion tasks for knowledge bases attract considerable attention, because knowledge bases are known to be far from complete, as discussed in (West et al., 2014) (Krompaß et al., 2015). Multi-relational graph embedding can help its completion and improve the performance of applications that use the graph. This is the reason why much work focuses on multi-relational graph embedding. Figure 1 shows an example of a multi-relational graph and a completion task.

In multi-relational graph embedding, reducing the number of parameters is an important problem in the era of big data. Many parameters are needed with tensor-factorization-based methods, such as Bayesian clustered tensor factorization (BCTF) (Sutskever et al., 2009), RESCAL (Nickel et al., 2011), and a neural tensor network (NTN) (Socher et al., 2013), where each relation has a dense matrix or tensors ($O\left(D^2\right)$ or more parameters, where $D$ is dimensionality of the space). Thus, TransE (Bordes et al., 2013) was proposed to reduce the number of parameters, to overcome this problem. In TransE, each entity is mapped to a point in Euclidean space and each relation is no more than a vector addition ($O\left(D\right)$ parameters), rather than a matrix operation. The successors to TransE, TransH (Wang et al., 2014) and TransD (Ji et al., 2016), also use only a small number of parameters. Some methods succeeded in reducing parameters using diagonal matrices instead of dense matrices: e.g. DISTMULT (Yang et al., 2015), ComplEx (Trouillon et al., 2016), HolE (through the Fourier transform) (Nickel et al., 2016), and ANALOGY (Liu et al., 2017). In these methods, all relations share one space for embedding, but each relation uses its own dissimilarity

criterion. The success of these methods implies that one common space underlies whole data, and each relation can be regarded as a dissimilarity criterion in the space.

Whereas these methods use distances or inner products in Euclidean space as dissimilarity criteria, recent work has shown that using non-Euclidean space can further reduce the number of parameters. One typical example of this is Poincaré Embedding (Nickel & Kiela, 2017) for hierarchical data, where a hyperbolic space is used as a space for embedding. Here, the tree structure of hierarchical data has good compatibility with the exponential growth of hyperbolic space. Recall the circumference with radius $R$ is given by $2\pi \sinh R (\approx 2\pi \exp R)$ in a hyperbolic plane. As a result, Poincaré embedding achieved good graph completion accuracy, even in low dimensionality such as 5 or 10. On the other hand, spheres (circumference: $2\pi \sin R$) are compatible with cyclic structures. Since Poincaré embedding, several methods have been proposed for single-relational graph embedding in non-Euclidean space (e.g. (Ganea et al., 2018b), (Nickel & Kiela, 2018)) and shown good results. The success of these methods suggests that the appropriate choice of a manifold (i.e., space) can retain low dimensionality, although these methods are limited to single-relational graph embedding.

According to the success of the TransE and its derivation and Poincaré embedding, it is reasonable in multi-relational graph embedding to assume the existence of a single structure compatible with a non-Euclidean manifold. For example, we can consider a single tree-like structure, which contains multiple hierarchical structures, where root selection gives multiple hierarchical structures from a single tree, which is compatible with hyperbolic spaces (See Figure 2). Therefore, embedding in a single shared non-Euclidean manifold with multiple dissimilarity criteria used in TransE is promising. Taking Poincaré embedding's success with low dimensionality into consideration, this method should work well (e.g., in graph completion tasks) with small number of parameters. This is the main idea of this paper.

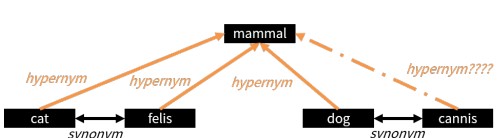

Figure 1: Multi-relational graph and its completion. There are five entities and two kinds of relation (hypernym and synonym). Graph completion refers to answering questions such as "is mammal a hypernym of cannis?"

Figure 2: Multiple hierarchical relations in a single tree. As this example shows, it is possible that multiple relations are given by multiple dissimilarity criteria in a single structure.

## 1.2 CONTRIBUTIONS

We propose a novel method, called *Riemannian TransE*, for multi-relation graph embedding using a non-Euclidean manifold. In Riemannian TransE, the relations share one non-Euclidean space and the entities are mapped to the space, whereas each relation has its own dissimilarity criterion based on the distance in the space. Specifically, the dissimilarity criteria in Riemannian TransE are similar to those in TransE (Bordes et al., 2013) based on vector addition, which is known to be effective. Unfortunately, we cannot straightforwardly use TransE's dissimilarity criteria. This is due to non-existence of a parallel vector field (See Figure 4), which is implicitly but essentially used in "vector addition." However, the parallel condition is not essential in TransE's idea. For example, hierarchical bottom to top relations should be regarded as attraction to the top in the hierarchy, which is not parallel but has an attractive point. Moreover, parallel vector fields can be regarded as a vector field attracted to a point at infinity. Therefore, we replace parallel vector fields in TransE by vector fields with an attractive point that are well-defined in Riemannian manifolds, and as a result, we obtain Riemannian TransE. Advantages of non-Euclidean spaces enable our Riemannian TransE to achieve good performance (e.g. in graph completion) with low-dimensional parameters. Riemannian TransE further exploits the advantages of TransE: that is, the method needs only $O(D)$ parameters for each relation. Numerical experiments on graph completion tasks show that with an appropriate choice of manifold, our method can improve the performance of multi-relational graph embedding with few parameters.

## 2 RELATED WORK

### 2.1 MULTI-RELATIONAL GRAPH EMBEDDING

Let $\mathcal{V}$ and $\mathcal{R}$ denote the entities and relations in a multi-relational graph, and let $\mathcal{T} \subset \mathcal{V} \times \mathcal{R} \times \mathcal{V}$ denote the triples in the graph. Multi-relational graph embedding refers to a pair of maps from $\mathcal{V}$ and $\mathcal{R}$ into $\mathcal{M}_e$ and $\mathcal{M}_r$, respectively. Particularly, learning multi-relational graph embedding refers to obtaining an appropriate pair of maps $v \mapsto p_v$ ($v \in \mathcal{V}, p_v \in \mathcal{M}_e$) and $r \mapsto w_r$ ($r \in \mathcal{R}, w_r \in \mathcal{M}_r$) from the triples $\mathcal{T}$. In this paper, we call $p_v$ the *planet* of entity $v$, $w_r$ the *launcher* of relation $r$, and $\mathcal{M}_e$ and $\mathcal{M}_r$ the *planet manifold* and *launcher manifold*, respectively. The quality of embedding is measured through a score function $f : (\mathcal{M}_e \times \mathcal{M}_e) \times \mathcal{M}_r \to \mathbb{R}$, which is designed by each method. Embedding is learned such that the value score function $f(p_h, p_t; w_r)$ will be low when $p_h, p_t; w_r \in \mathcal{T}$ and high when $p_h, p_t; w_r \notin \mathcal{T}$. For specific loss functions designed from the score function, see Subsection 2.3. We interpret the score function of multi-relational graph embedding as dissimilarity in a manifold, which we call a *satellite* manifold $\mathcal{M}_s$. We rewrite the score function $f$ in multi-relational graph embedding using two maps $\mathscr{H}, \mathscr{T} : \mathcal{M}_e \times \mathcal{M}_r \to \mathcal{M}_s$ and the dissimilarity measure function $\mathcal{D} : \mathcal{M}_s \times \mathcal{M}_s \to \mathbb{R}$ as follows:

$$f(p_h, p_t; w_r) \coloneqq \mathcal{D}\left(s_{h;r}^{\mathrm{H}}, s_{t;r}^{\mathrm{T}}\right), \text{ where } s_{h;r}^{\mathrm{H}} = \mathscr{H}(p_h; w_r), s_{t;r}^{\mathrm{T}} = \mathscr{T}(p_t; w_r). \tag{1}$$

We call $\mathscr{H}$ and $\mathscr{T}$ the *head* and *tail launch map*, respectively, and call $s_{v;r}^{\mathrm{H}}$ and $s_{v;r}^{\mathrm{T}}$ the *head* and *tail satellite* of entity $v$ (or of planet $p_v$) with respect to relation $r$. The idea of this formulation is embedding in one shared space with multiple dissimilarity criteria. Specifically, each entity has only one planet and their satellite pairs give multiple dissimilarity criteria, each of which corresponds to a relation. In other words, all of the relations shares one space and the planets in it, and the differences among the relations are reduced to the difference of their launcher maps and the satellites given by them. We regard the planets as the embeddings of the entities, whereas dissimilarity between entities with respect to a relation is evaluated through their satellites which correspond to the relation.

A simple example of this is TransE (Bordes et al., 2013), where all of the planets, satellites, and launchers share the same Euclidean space, i.e. $\mathcal{M}_e = \mathcal{M}_s = \mathcal{M}_r = \mathbb{R}^D$, the launch maps are given by vector addition as $\mathscr{H}(p; w) = p + w$ and $\mathscr{T}(p; w) = p$, and the distance in a norm space— i.e. the norm of the difference—is used as a dissimilarity criterion i.e. $\mathcal{D}(s^{\mathrm{H}}, s^{\mathrm{T}}) = \|s^{\mathrm{T}} - s^{\mathrm{H}}\|$ (the L1 or L2 norm is often used in practice). See Figure 5 (left). As Nguyen (2017) suggested, one can associate the idea of representing relations as vector additions with the fact that we can find a relation through a substraction operator in Word2Vec Mikolov et al. (2013). That is, we can find relations such as $p_{\mathrm{France}} - p_{\mathrm{Paris}} \approx p_{\mathrm{Italy}} - p_{\mathrm{Rome}}$ in Word2Vec. As explained above, TransE is based on the distance between satellites, and each satellite is given by simple vector addition. Regardless of this simplicity, the performance of TransE has been exemplified in review papers (Nickel et al., 2016) (Nguyen, 2017). Indeed, the addition operation in a linear space is essential in the launcher map, and hence TransE can easily be extended to a Lie group, which is a manifold equipped with an addition operator, as suggested in Ebisu & Ichise (2017). Some methods, such as TransH (Wang et al., 2014), TransR (Lin et al., 2015), and TransD (Ji et al., 2016), also use a norm in linear space as a dissimilarity measure, integrating a linear map into a latent space.

Another simple example is RESCAL (Nickel et al., 2011), which uses the negative inner product as a dissimilarity measure. In RESCAL, the launcher of relation $r$ is a matrix $\boldsymbol{W} \in \mathcal{M}_r = \mathbb{R}^{D \times D}$, the launch maps are given by a linear map, i.e. $\mathscr{H}(p; (\boldsymbol{W}, w)) = \boldsymbol{W}p$ and $\mathscr{T}(p; (\boldsymbol{W}, w)) = p$, and the dissimilarity measure is the negative inner product $\mathcal{D}(s^{\mathrm{H}}, s^{\mathrm{T}}) = -(s^{\mathrm{H}})^{\top} s^{\mathrm{T}}$. Other methods are also based on the (negative) inner product dissimilarity: e.g., DISTMULT (Yang et al., 2015), ComplEx (Trouillon et al., 2016), HolE (through the Fourier transform) (Nickel et al., 2016), and ANALOGY (Liu et al., 2017). Table 1 shows score functions of these methods.

Whereas some methods are based on a neural network (e.g., the neural tensor network (Socher et al., 2013) and ConvE (Dettmers et al., 2017)), their score function consists of linear operations and element-wise nonlinear functions.

### 2.2 GRAPH EMBEDDING IN NON-EUCLIDEAN SPACE

Graph embedding using non-Euclidean space has attracted considerable attention, recently. Specifically, embedding methods using hyperbolic space have achieved outstanding results (Nickel & Kiela,

Table 1: Score Functions. The launcher $\boldsymbol{w}_r$ of $r$ determines the dissimilarity criterion of $r$ through satellites. In this table, the dimensionality is set so that the (real) dimensionality of the planets is $D$. † denotes conjugate transpose. $\mathfrak{F}$ denotes the discrete Fourier Transform. The interpretation here of HolE is given by Liu et al. (2017) and Hayashi & Shimbo (2017).

| Model | Planets Launchers | Head satellites $s^{\mathrm{H}}_{h;r}$ Tail satellites $s^{\mathrm{T}}_{t;r}$ | Dissimilarity # parameters |
|---|---|---|---|
| **Riemannian TransE** **This paper** | $p_v \in \mathcal{M}$ ($D$-dim.) $w_r = (\ell_r, p_r) \in \mathbb{R} \times \mathcal{M}$ | $\mathfrak{m}_{[\ell_r]_+, p_r}(p_h) \in \mathcal{M}$ (See (6)) $\mathfrak{m}_{[-\ell_r]_+, p_r}(p_t) \in \mathcal{M}$ (See (6)) | $\Delta\left(s^{\mathrm{H}}_{h;r}, s^{\mathrm{T}}_{t;r}\right)$ $D\,|\mathcal{V}| + (D+1)\,|\mathcal{R}|$ |
| TransE Bordes et al. (2013) | $\boldsymbol{p}_v \in \mathbb{R}^D$ $\boldsymbol{w}_r \in \mathbb{R}^D$ | $\boldsymbol{p}_h + \boldsymbol{w}_r \in \mathbb{R}^D$ $\boldsymbol{p}_t \in \mathbb{R}^D$ | $\left\| s^{\mathrm{T}}_{t;r} - s^{\mathrm{H}}_{h;r} \right\|$ $D\,|\mathcal{V}| + D\,|\mathcal{R}|$ |
| TransH Wang et al. (2014) | $\boldsymbol{p}_v \in \mathbb{R}^D$ $(\boldsymbol{w}_r, \boldsymbol{w}_r^{\mathrm{pr}}) \in \mathbb{R}^D \times \mathbb{R}^D$ | $\left[\mathbf{I} - \boldsymbol{w}_r^{\mathrm{pr}}\boldsymbol{w}_r^{\mathrm{pr}\top}\right]\boldsymbol{p}_h + \boldsymbol{w}_r \in \mathbb{R}^D$ $\left[\mathbf{I} - \boldsymbol{w}_r^{\mathrm{pr}}\boldsymbol{w}_r^{\mathrm{pr}\top}\right]\boldsymbol{p}_t \in \mathbb{R}^D$ | $\left\| s^{\mathrm{T}}_{t;r} - s^{\mathrm{H}}_{h;r} \right\|$ $D\,|\mathcal{V}| + 2D\,|\mathcal{R}|$ |
| TransR Lin et al. (2015) | $\boldsymbol{p}_v \in \mathbb{R}^D$ $(\boldsymbol{W}_r, \boldsymbol{w}_r) \in \mathbb{R}^{D \times \tilde{D}} \times \mathbb{R}^{\tilde{D}}$ | $\boldsymbol{W}_r\boldsymbol{p}_h + \boldsymbol{w}_r \in \mathbb{R}^{\tilde{D}}$ $\boldsymbol{W}_r\boldsymbol{p}_t \in \mathbb{R}^{\tilde{D}}$ | $\left\| s^{\mathrm{T}}_{t;r} - s^{\mathrm{H}}_{h;r} \right\|$ $D\,|\mathcal{V}| + \left(D\tilde{D} + \tilde{D}\right)|\mathcal{R}|$ |
| TransD Ji et al. (2016) | $(\boldsymbol{p}_v, \boldsymbol{p}_v^{pr}) \in \mathbb{R}^{D/2} \times \mathbb{R}^{D/2}$ $(\boldsymbol{w}_r, \boldsymbol{w}_r^{\mathrm{pr}}) \in \mathbb{R}^{\tilde{D}} \times \mathbb{R}^{\tilde{D}}$ | $\left[\mathbf{I} + \boldsymbol{w}_r^{\mathrm{pr}}\boldsymbol{p}_h^{\mathrm{pr}\top}\right]\boldsymbol{p}_h + \boldsymbol{w}_r \in \mathbb{R}^{\tilde{D}}$ $\left[\mathbf{I} + \boldsymbol{w}_r^{\mathrm{pr}}\boldsymbol{p}_t^{\mathrm{pr}\top}\right]\boldsymbol{p}_t \in \mathbb{R}^{\tilde{D}}$ | $\left\| s^{\mathrm{T}}_{t;r} - s^{\mathrm{H}}_{h;r} \right\|$ $D\,|\mathcal{V}| + 2\tilde{D}\,|\mathcal{R}|$ |
| RESCAL Nickel et al. (2011) | $\boldsymbol{p}_v \in \mathbb{R}^D$ $\boldsymbol{W} \in \mathbb{R}^{D \times D}$ | $\boldsymbol{W}_r\boldsymbol{p}_h \in \mathbb{R}^D$ $\boldsymbol{p}_t \in \mathbb{R}^D$ | $-s^{\mathrm{H}}_{h;r}{}^\top s^{\mathrm{T}}_{t;r}$ $D\,|\mathcal{V}| + D^2\,|\mathcal{R}|$ |
| DISTMULT Yang et al. (2015) | $\boldsymbol{p}_v \in \mathbb{R}^D$ $\boldsymbol{w}_r \in \mathbb{R}^D$ | $\mathrm{diag}\{\boldsymbol{w}_r\}\,\boldsymbol{p}_h \in \mathbb{R}^D$ $\boldsymbol{p}_t \in \mathbb{R}^D$ | $-s^{\mathrm{H}}_{h;r}{}^\top s^{\mathrm{T}}_{t;r}$ $D\,|\mathcal{V}| + D\,|\mathcal{R}|$ |
| ComplEx Trouillon et al. (2016) | $\boldsymbol{p}_v \in \mathbb{C}^{D/2}$ $\boldsymbol{w}_r \in \mathbb{C}^{D/2}$ | $\mathrm{diag}\{\boldsymbol{w}_r\}\,\boldsymbol{p}_h \in \mathbb{C}^{D/2}$ $\boldsymbol{p}_t \in \mathbb{C}^{D/2}$ | $-\mathrm{Re}\left(s^{\mathrm{H}}_{h;r}{}^\dagger s^{\mathrm{T}}_{t;r}\right)$ $D\,|\mathcal{V}| + D\,|\mathcal{R}|$ |
| HolE Nickel et al. (2016) | $\boldsymbol{p}_v \in \mathbb{R}^D$ $\boldsymbol{w}_r \in \mathbb{R}^D$ | $\mathfrak{F}(\boldsymbol{p}_h) \in \mathbb{C}^D$ $\mathrm{diag}\{\mathfrak{F}(\boldsymbol{w}_r)\}\,\mathfrak{F}(\boldsymbol{p}_t) \in \mathbb{C}^D$ | $-\mathrm{Re}\left(s^{\mathrm{H}}_{h;r}{}^\dagger s^{\mathrm{T}}_{t;r}\right)$ $D\,|\mathcal{V}| + D\,|\mathcal{R}|$ |
| ANALOGY Liu et al. (2017) | $\left(\boldsymbol{p}_v^{\mathbb{C}}, \boldsymbol{p}_v^{\mathbb{R}}\right) \in \mathbb{C}^{D/4} \times \mathbb{R}^{D/2}$ $\left(\boldsymbol{w}_r^{\mathbb{C}}, \boldsymbol{w}_r^{\mathbb{R}}\right) \in \mathbb{C}^{D/4} \times \mathbb{R}^{D/2}$ | $\mathrm{diag}\{\boldsymbol{w}_r\}\,\boldsymbol{p}_h \in \mathbb{C}^{\frac{3}{4}D}$ $\boldsymbol{p}_t \in \mathbb{C}^{\frac{3}{4}D}$ | $-\mathrm{Re}\left(s^{\mathrm{H}}_{h;r}{}^\dagger s^{\mathrm{T}}_{t;r}\right)$ $D\,|\mathcal{V}| + D\,|\mathcal{R}|$ |

2017) (Ganea et al., 2018b) (Nickel & Kiela, 2018). With these methods, each node in the graph is mapped to a point in hyperbolic space and the dissimilarity is measured by a distance function in the space. Although these methods exploit the advantages of non-Euclidean space, specifically those of a negative curvature space, they focus on single- rather than multi-relational graph embedding.

By contrast, TransE has been extended to an embedding method in a Lie group—that is, a manifold with the structure of a group (Ebisu & Ichise, 2017). As such, the regularization problem in TransE is avoided by using torus, which can be regarded as a Lie group. Although this extension to TransE deals with multi-relational embedding, it cannot be applied to all manifolds. This is because not all manifolds have the structure of a Lie group. Indeed, we cannot regard a hyperbolic space (if $D \neq 1$) or a sphere (if $D \neq 1, 3$) as a Lie group.

## 2.3 LOSS FUNCTION

We can simply design a loss function on the basis of the negative log likelihood of a Bernoulli model as follows:

$$\mathcal{L}\left(\{p_v\}_{v \in \mathcal{V}}, \{w_r\}_{r \in \mathcal{R}}\right) := -\sum_{(h,r,t) \in \mathcal{T}} \log\left(\sigma\left(f\left(p_h, p_t; w_r\right)\right)\right) - \sum_{(h',r',t') \in \mathcal{T}^{\mathrm{c}}} \log\left(1 - \sigma\left(f\left(p_{h'}, p_{t'}; w_{r'}\right)\right)\right),$$

(2)

where $\mathcal{T}^{\mathrm{c}} := (\mathcal{V} \times \mathcal{R} \times \mathcal{V}) \setminus \mathcal{T}$ and $\sigma : \mathbb{R} \to [0, 1]$ is a sigmoid function. However, this loss function needs evaluation of the score function for all negative triplets $(\mathcal{V} \times \mathcal{R} \times \mathcal{V}) \setminus \mathcal{T}$. To avoid this, most methods (e.g., TransE) use the following margin-based loss function:

$$\mathcal{L}\left(\{p_v\}_{v \in \mathcal{V}}, \{w_r\}_{r \in \mathcal{R}}\right) := \sum_{(h,h',r,t',t) \in \mathcal{Q}} \left[\delta + f\left(p_h, p_t; w_r\right) - f\left(p_{h'}, p_{t'}; w_r\right)\right]_+,$$

(3)

where $\mathcal{Q}$ is the set of the triples with its corrupted head and tail. That is,

$$\mathcal{Q} := \{(h, h', r, t', t) \in \mathcal{V} \times \mathcal{V} \times \mathcal{R} \times \mathcal{V} \times \mathcal{V} \mid [(h, r, t) \in \mathcal{T}] \wedge [(h' = h) \vee (t' = t)]\}, \quad (4)$$

where $\delta \in \mathbb{R}_{\geq 0}$ is the margin hyperparameter, and $[\cdot]_+$ denotes the negative value clipping—i.e. for all $x \in \mathbb{R}$, $[x]_+ := \max(x, 0)$. We use this loss function throughout this paper.

# 3 RIEMANNIAN TRANSE

In this section, we formulate Riemannian TransE exploiting the advantages of TransE in non-Euclidean manifolds. Firstly, we give a brief introduction of Riemannian geometry. Secondly, we explain the difficulty in application of TransE in non-Euclidean manifolds. Lastly, we formulate Riemannian TransE.

## 3.1 RIEMANNIAN MANIFOLDS AND OPERATIONS

Let $(\mathcal{M}, \mathfrak{g})$ be a Riemannian manifold with metric $\mathfrak{g}$. We denote the tangent and cotangent space of $\mathcal{M}$ on $p$ by $\mathfrak{T}_p\mathcal{M}$ and $\mathfrak{T}_p^*\mathcal{M}$, respectively, and we denote the collection of all smooth vector fields on $\mathcal{M}$ by $\mathfrak{X}(\mathcal{M})$. Let $\nabla : \mathfrak{X}(\mathcal{M}) \times \mathfrak{X}(\mathcal{M}) \ni (X, Y) \mapsto \nabla_X Y \in \mathfrak{X}(\mathcal{M})$ denote the Levi–Civita connection, the unique metric-preserving torsion-free affine connection. A smooth curve $\gamma : (-\epsilon, \epsilon) \to \mathcal{M}$ is a *geodesic* when $\nabla_{\dot{\gamma}}\dot{\gamma} = 0$ on curve $\gamma$, where $\dot{\gamma}$ is the differential of curve $\gamma$. Geodesics are generalizations of straight lines, in the sense that they are constant speed curves that are locally distance-minimizing. We define the exponential map $\mathrm{Exp}_p$, which moves point $p \in \mathcal{M}$ towards a vector by the magnitude of the vector. In this sense, the exponential map is regarded as an extension of vector addition in a Riemannian manifold. Figure 3 shows an intuitive example of an exponential map on a sphere. Let $\gamma_v$ $(v \in \mathfrak{T}_p\mathcal{M})$ denote the geodesic that satisfies $\dot{\gamma}_v(0) = v$. The exponential map $\mathrm{Exp}_p : \mathfrak{T}_p\mathcal{M} \to \mathcal{M}$ is given by $\mathrm{Exp}_p(v) := \gamma_v(1)$. We define the logarithmic map $\mathrm{Log}_p : \mathcal{M} \to \mathfrak{T}_p\mathcal{M}$ as the inverse of the exponential map. Note that the exponential map is not always bijective, and we have to limit the domain of the exponential and logarithmic map appropriately, while some manifolds, such as Euclidean and hyperbolic space, do not suffer from this problem.

## 3.2 DIFFICULTIES IN RIEMANNIAN MANIFOLDS

In TransE, a single vector $\boldsymbol{w}_r$ determines the head and tail launch maps $\mathscr{H}, \mathscr{T}$ as a transform: $\mathbb{R}^D \to \mathbb{R}^D$. In fact, these launch maps are given by vector addition. Note that this constitution of the launcher maps implicitly but essentially uses the fact that a vector is identified with a parallel vector field in Euclidean space. Specifically, a vector $\boldsymbol{w}$ determines a parallel vector field, denoted by $W_r$ here, which gives a tangent vector $[W_r]_p \in \mathfrak{T}_p\mathbb{R}^D$ on every point $p \in \mathbb{R}^D$, and each tangent vector determines the exponential map $\mathrm{Exp}_p([W_r]_p)$ at $p$, which is used as a launch map in TransE. However, because there is no parallel vector field in non-zero curvature spaces, we cannot apply TransE straightforwardly in non-zero curvature spaces. Thus, extention of TransE in non-Euclidean space non-trivial. This is the difficulty in Riemannian Manifolds.

## 3.3 FORMULATION OF RIEMANNIAN TRANSE

As we have explained in Introduction, our idea is replacing parallel vector fields in TransE by vector fields attracted to a point. Specifically, we obtain the *Riemannian TransE* as an extension of TransE, replacing the launchers $\boldsymbol{w}_r \in \mathbb{R}^D$ in TransE by pairs $w_r = (\ell_r, p_r) \in \mathbb{R} \times \mathcal{M}$ of a scalar value and point, indicating the length and destination of the satellites' move, respectively. We call $p_r$ the *attraction point* of relation $r$. In other words, we replace parallel vector field $W_r = \boldsymbol{w}_r$ in TransE by $\ell_r \frac{\mathrm{Log}_q(p)}{\left\|\mathrm{Log}_q(p)\right\|_q}$. Note that, we use a fixed manifold $\mathcal{M}_\mathrm{e} = \mathcal{M}$ for entity embedding and use direct product manifold $\mathcal{M}_\mathrm{r} = \mathbb{R} \times \mathcal{M}$ for relation embedding.

However, the extension still has arbitrariness. For instance, we could launch the tail satellite instead of the head satellite in TransE; in other words, the following launching map also gives us a score function equivalent to that of the original TransE: $\mathscr{H}(\boldsymbol{p}; \boldsymbol{w}) = \boldsymbol{p}$ and $\mathscr{T}(\boldsymbol{p}; \boldsymbol{w}) = \boldsymbol{p} - \boldsymbol{w}$ (Figure 5 center). On the other hand, the score function depends on whether we move the head or tail satellites

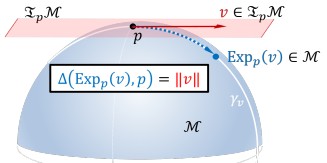

Figure 3: Tangent space and exponential map. The exponential map moves the point $p \in \mathcal{M}$ along a geodesic (the white line) that tangent to $v \in \mathfrak{T}_p\mathcal{M}$.

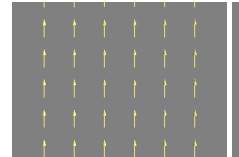 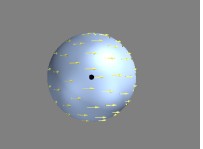 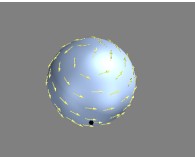

Figure 4: Parallel vector field in a sphere. The left figure shows a parallel vector field in a plane. In a sphere, there is no parallel vector field. Even if a vector field seems parallel from one view (center), it turns out to be not parallel (right)

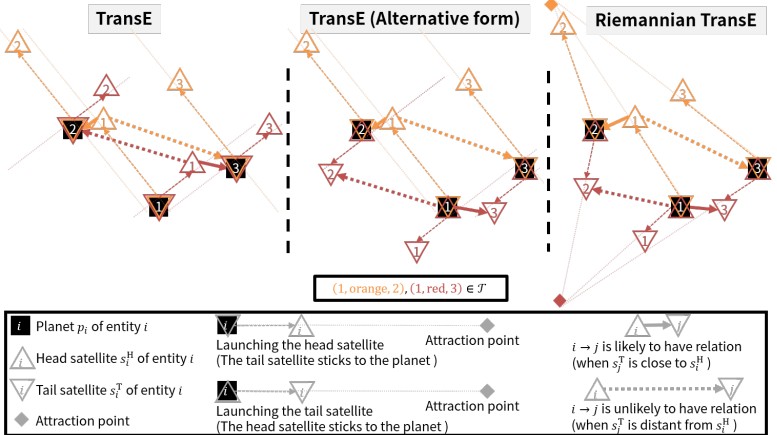

Figure 5: Difference between TransE and Riemannian TransE. In these examples, the number $|\mathcal{V}|$ of entities is three (1, 2, 3) and the number $|\mathcal{R}|$ of relations is two (red and orange), with triples (1, orange, 2) and (1, red, 3). Hence, these models learn that the orange head satellite of Entity 1 is close to the orange tail satellite of Entity 2 and the red head satellite of Entity 1 is close to the red tail satellite of Entity 3. In addition, the distance of the other pair of satellites should be long in the representation learned by each method. The figure on the left shows the original formulation of TransE, where the satellites are given by vector addition. In other words, the satellites are given by a move towards a point at infinity from the planet. The center figure shows an alternative formulation of TransE, which is equivalent to the original TransE. Here, the tail satellites are launched and the head satellites are fixed in the red relation. In Riemannian TransE in the figure on the right, the vector additions are replaced by a move towards a (finite) point.

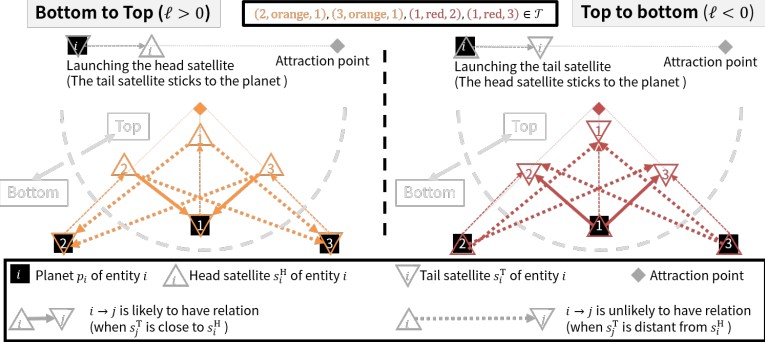

Figure 6: Relation of the sign for $\ell$. If $\ell$ is positive (e.g. the orange relation), the relation runs from low (e.g. Entity 2 and 3) to high hierarchy (e.g. Entity 1), and vice versa (e.g. the red relation).

in our case, where the attraction points are not at infinity. With hierarchical data, an entity at a higher hierarchy has many related entities in a lower hierarchy. Therefore, it is best to always launch the satellites of "children," the entities in a lower hierarchy, toward their parent. Hence, we move the

head satellites when $\ell_r > 0$ and fix the tail satellites, and vice versa when $\ell_r < 0$; specifically, we move the head satellites by length $\lambda = [\ell_r]_+$ and move the tail satellites by length $\lambda = [-\ell_r]_+$. Thus, bottom-to-top relation cases correspond to $\ell_r > 0$ (Figure 6, left), and top-to-bottom relation cases correspond to $\ell_r < 0$ (Figure 6, right). Another problem pertains to launching the satellites near the attraction point. If $\lambda > \Delta(p_r, p_v)$, the naive rule causes overrun. In this case, we simply clip the move and set the satellite in the place of $p_r$.

We turn now to the score function of Riemannian TransE. The score function $f : (\mathcal{M} \times \mathcal{M}) \times (\mathbb{R}, \mathcal{M}) \to \mathbb{R}$ in Riemannian TransE is given as follows:

$$f(p_h, p_t; (\ell_r, p_r)) \coloneqq \Delta\left(s_{h;r}^{\mathrm{H}}, s_{t;r}^{\mathrm{T}}\right), \text{ where } \begin{cases} s_{h;r}^{\mathrm{H}} \coloneqq \mathscr{H}(p_h; (\ell_r, p_r)) \coloneqq \underset{[\ell_r]_+, p_r}{\mathfrak{m}}(p_h), \\ s_{t;r}^{\mathrm{T}} \coloneqq \mathscr{T}(p_t; (\ell_r, p_r)) \coloneqq \underset{[-\ell_r]_+, p_r}{\mathfrak{m}}(p_t), \end{cases} \tag{5}$$

where transform $\underset{\lambda, p}{\mathfrak{m}} : \mathcal{M} \to \mathcal{M}$ denotes a move, defined as follows:

$$\underset{\lambda, p}{\mathfrak{m}}(q) \coloneqq \mathrm{Exp}_p\left([\Delta(q, p) - \lambda]_+ \frac{\mathrm{Log}_p(q)}{\|\mathrm{Log}_p(q)\|_p}\right). \tag{6}$$

Here, note that $\underset{\ell, p}{\mathfrak{m}}(q)$ is on the geodesic that passes through $p$ and $q$. Figure 5 (right) shows the Riemannian TransE model. If $\mathcal{M} = \mathbb{R}^D$ and the attraction points are at infinity, the score function is equivalent to that of TransE (without the sphere constraint). Although the exponential map and logarithmic map in closed form are required to implement Riemannian TransE, we can obtain them when the manifold $\mathcal{M}$ is a sphere $\mathbb{S}^D$ (positive curvature), Euclidean space $\mathbb{R}^D$ (zero curvature), and hyperbolic space $\mathbb{H}^D$ (negative curvature), or a direct product of them. These are practically sufficient. Also note that the computation costs of these maps are $O(D)$, which is small enough.

### 3.4 OPTIMIZATION

In typical cases, the number of entities is very large. Therefore, stochastic gradient methods are effective for optimization. Although we can directly apply stochastic gradient methods of Euclidean space or the natural gradient method (Amari, 1998), Riemannian gradient methods (e.g. (Zhang & Sra, 2016) (Zhang et al., 2016)) work better for non-Euclidean embedding (Enokida et al., 2018). In this paper, we use stocastic Riemannian sub gradient methods Zhang & Sra (2016) with norm clipping (See Appendix). Note that in spheres or hyperbolic spaces, the computation costs of the gradient is $O(D)$, which is as small as TransE.

Table 2: Triple classification performance. **Bold**: Top 1, *Italic*: Top 3.

| Dataset | WN11 | | | | | FB13 | | | | |
|---|---|---|---|---|---|---|---|---|---|---|
| Dim. | 8 | 16 | 32 | 64 | 128 | 8 | 16 | 32 | 64 | 128 |
| **Hyperbolic TransE** | 64.74 | 66.51 | 67.78 | 67.92 | 67.87 | *80.05* | *78.06* | **77.53** | **84.65** | **84.67** |
| **PHyperbolic TransE** | 68.51 | 72.88 | 74.70 | *75.83* | *77.03* | *78.42* | 77.06 | *77.39* | *77.74* | *78.53* |
| **Spherical TransE** | **82.07** | **83.11** | **82.99** | **83.13** | **83.30** | 64.45 | 63.38 | 64.69 | 70.07 | 69.74 |
| **PSpherical TransE** | *80.73* | *81.37* | *77.12* | 69.05 | 63.42 | 71.26 | 71.34 | 71.23 | 73.03 | 74.83 |
| **Euclidean TransE** | 72.66 | 73.99 | *75.27* | *76.69* | *77.04* | **81.84** | *80.03* | 75.44 | 76.99 | 77.52 |
| TransE | 60.94 | 64.63 | 63.20 | 61.92 | 58.46 | 67.60 | 68.29 | 68.86 | 75.68 | 74.92 |
| TransE (unconstraint) | 67.55 | 66.18 | 64.07 | 63.23 | 61.51 | 76.44 | **80.22** | *77.24* | 76.01 | 75.59 |
| TorusE | 62.34 | 62.78 | 63.33 | 63.19 | 63.45 | 61.51 | 58.04 | 63.06 | 60.31 | 58.14 |
| TransH | *77.55* | *75.44* | 70.03 | 65.46 | 63.75 | 71.07 | 75.25 | 76.89 | *78.32* | *80.32* |
| TransR | 52.58 | 53.13 | 55.30 | 53.10 | 55.80 | 52.41 | 52.41 | 51.65 | 51.87 | 52.38 |
| TransD | 53.43 | 54.61 | 55.76 | 63.32 | 61.59 | 55.02 | 56.68 | 53.69 | 56.28 | 56.02 |
| RESCAL | 60.36 | 57.65 | 56.85 | 56.62 | 57.62 | 74.28 | 70.17 | 67.88 | 65.90 | 63.20 |
| DistMult | 61.05 | 61.01 | 58.97 | 57.11 | 55.85 | 64.54 | 65.04 | 63.32 | 59.77 | 54.76 |
| ComplEx | 62.63 | 62.47 | 57.91 | 56.02 | 53.47 | 70.07 | 72.36 | 71.11 | 67.36 | 64.49 |
| HolE | 53.01 | 53.03 | 51.19 | 52.62 | 53.09 | 58.12 | 62.13 | 61.35 | 60.74 | 54.61 |
| Analogy | 63.60 | 59.24 | 58.55 | 57.51 | 57.00 | 66.38 | 66.18 | 64.54 | 59.48 | 55.26 |

## 4 EXPERIMENTS

**Evaluation Tasks** We evaluated the performance of our method for a triple classification task (Socher et al., 2013) on real knowledge base datasets. The triple classification task involved predict-

ing whether a triple in the test data is correct. We label a triple positive when $f\left(p_h, p_t; (\ell_r, p_r)\right) > \theta_r$, and vice versa. Here, $\theta_r \in \mathbb{R}_{\geq 0}$ denotes the threshold for each relation $r$, which is determined by the accuracy of the validation set. We evaluated the accuracy of classification with the FB13 and WN11 datasets (Socher et al., 2013). Although we do not report the results of link prediction tasks (Bordes et al., 2013) here because there are many evaluation criteria for the task, which makes it difficult to interpret the results, we report the results in Appendix.

**Manifolds in Riemannian TransE**    To evaluate the dependency of performance for Riemannian TransE, we compared Riemannian TransE using the following five kinds of manifolds: Euclidean space $\mathbb{R}^D$ (Euclidean TransE), hyperbolic space $\mathbb{H}^D$ (Hyperbolic TransE), a sphere $\mathbb{S}^D$ (Spherical TransE), the direct product $\mathbb{H}^4 \times \mathbb{H}^4 \times \cdots \times \mathbb{H}^4$ of hyperbolic space (PHyperbolic TransE), and the direct product $\mathbb{S}^4 \times \mathbb{S}^4 \times \cdots \times \mathbb{S}^4$ of a sphere (PSpherical TransE).

**Baselines and Implementation**    We compared our method with the following baselines: RESCAL (Nickel et al., 2011), TransE (Bordes et al., 2013), TransH (Wang et al., 2014), TransR (Lin et al., 2015), TransD (Ji et al., 2016), TorusE Ebisu & Ichise (2017), RESCAL (Nickel et al., 2011), DISTMULT (Yang et al., 2015), HolE (Nickel et al., 2016), ComplEx (Trouillon et al., 2016) and Analogy (Liu et al., 2017). We used implementations of these methods on the basis of OpenKE `http://openke.thunlp.org/static/index.html`, and we used the evaluation scripts there. Note that we compensated for some missing constraints (for example, in TransR and TransD) and regularizers (for example, in DISTMULT and Analogy) in OpenKE. We also found that omitting the constraint of the entity planets onto the sphere in TransE gave much better results in our setting, so we also provide these unconstrained results (UnconstraintTransE). We determined the hyperparameters by following each paper. For details, see the Appendix.

**Results**    Table 2 shows the results for the triple classification task in each dimensionality. In WN11, the sphere-based Riemannian TransEs achieved good accuracy. The accuracy did not degrade dramatically even with low dimensionality. On the other hand, in FB13, the hyperbolic-space-based Riemannian TransEs was more accurate than other methods. Moreover for each dimensionality, these results with the proposed Riemannian TransE were at least comparable to those of the baselines. The accuracy of Euclidean-space-based methods (e.g. the original TransE, and Euclidean TransE) are between that of the sphere-based Riemannian TransEs and that of the hyperbolic-space-based Riemannian TransEs in most cases. Note that these results are compatible with the curvature of each space (i.e. Sphere: positive, Euclidean space: 0, a hyperbolic space: negative). Note that Euclidean methods are sometimes better than non-Euclidean methods. In Appendix, we also report the triple classification task results in FB15k, where Euclidean TransE as well as baseline methods outperformed Riemannian TransE did not always outperform the baseline methods. In summary, positive curvature spaces were good in WN11 and negative curvature spaces were good in FB13, and zero curvature spaces were good in FB15k. These results show that Riemannian TransE can attain good accuracy with small dimensionality provided that an appropriate manifold is selected. What determines the appropriate manifold? Spheres are compatible with cyclic structure and hyperbolic spaces are compatible with tree-like structure. One possible explanation is that WN11 has cyclic structure and FB13 has tree-like structure and the structure of FB15k is between them. However, further discussion remains future work.

## 5    CONCLUSION AND FUTURE WORK

We proposed Riemannian TransE, a novel framework for multi-relational graph embedding, by extending TransE to a Riemannian TransE. Numerical experiments showed that Riemannian TransE outperforms baseline methods in low dimensionality, although its performance depends significantly on the choice of manifold. Hence, future research shall clarify which manifolds work well with particular kinds of data, and develop a methodology for choosing the appropriate manifold. This is important work not only for graph completion tasks but also for furthering our understanding of the global characteristics of a graph. In other words, observing which manifold is effective can help us to understand the global "behavior" of a graph. Other important work involves using "subspaces" in non-Euclidean space. Although the notion of a subspace in a non-Euclidean manifold is nontrivial, it may be that our method offers advantages over TransH and TransD, which exploit linear subspaces.

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

## A   OPTIMIZATION

In this paper, we use the following simple (projected) stochastic (Riemannian) (sub-) gradient methods Zhang & Sra (2016)

$$\theta_{(\tau+1)} \leftarrow \mathrm{Exp}_{\theta_{(\tau)}} \left( -\eta \tilde{\nabla}_{(\tau)} \right), \tag{7}$$

where $\theta_{(\tau)} \in \mathcal{M}^{|\mathcal{V}|} \times (\mathbb{R} \times \mathcal{M})^{|\mathcal{R}|}$ denotes the parameter in the $\tau$-th step, $\eta \in \mathbb{R}_{\geq 0}$ is the learning rate, and $\tilde{\nabla}_{(\tau)} \in \mathfrak{T}_{\theta_{(\tau)}} \left( \mathcal{M}^{|\mathcal{V}|} \times (\mathbb{R} \times \mathcal{M})^{|\mathcal{R}|} \right)$ is a stochastic gradient that satisfies $\mathbb{E}\left[ \tilde{\nabla}_{(\tau)} \right] = \mathrm{grad}\mathcal{L}\left( \theta_{(\tau)} \right) = \sharp \left( \mathrm{d}\mathcal{L}\left( \theta_{(\tau)} \right) \right)$. Recall that $\sharp$ denotes index raising. Specifically, we use the following stochastic loss function based on the mini-batch method:

$$\tilde{\mathcal{L}}\left( \theta_{(\tau)} \right) \coloneqq \sum_{(h,h',r,t',t)\in\mathcal{Q}'} \left[ \delta + f\left( p_h, p_t; w_r \right) - f\left( p_{h'}, p_{t'}; w_r \right) \right]_+, \tag{8}$$

where the stochastic quintet set $\mathcal{Q}'_{(\tau)} \subset \mathcal{Q}$ is a set of uniform-distributed random variables on $\mathcal{Q}$. $\Delta \left( s_r^{\mathrm{H}}\left( p_h \right), s_r^{\mathrm{T}}\left( p_t \right) \right)$. We obtain a stochastic gradient as follows:

$$\tilde{\nabla}_{(\tau)}^\flat = \mathrm{d}\tilde{\mathcal{L}}\left( \theta_{(\tau)} \right) = \frac{\partial}{\partial \boldsymbol{\theta}^\top} \tilde{\mathcal{L}}\left( \theta_{(\tau)} \right) \mathrm{d}\boldsymbol{\theta}, \quad \tilde{\nabla}_{(\tau)} = \sharp \left( \tilde{\nabla}_{(\tau)}^\flat \right) \tag{9}$$

where $\boldsymbol{\theta}$ is a local coordinate representation of $\theta$. We obtain $\tilde{\nabla}_{(\tau)}^\flat$ easily using an automatic differentiation framework. Algorithm 1 shows the learning algorithm for Riemannian TransE. In the experiments, we applied norm clipping such that the norm of a stochastic gradient is smaller than 1.

---

**Algorithm 1** Learning Riemannian TransE

---

**for** $\tau = 1, 2, \ldots$ **do**
    Sample $\mathcal{Q}'_{(\tau)}$ from uniform distribution on $\mathcal{Q}$.
    $\tilde{\nabla}_{(\tau)} \leftarrow \sharp \left( \frac{\partial}{\partial \boldsymbol{\theta}^\top} \sum_{(h,h',r,t',t)\in\mathcal{Q}'} \left[ \delta + f\left( p_h, p_t; w_r \right) - f\left( p_{h'}, p_{t'}; w_r \right) \right]_+ \right)$
    $\theta_{(\tau+1)} \leftarrow \mathrm{Exp}_{\theta_{(\tau)}} \left( -\eta_{(\tau)} \tilde{\nabla}_{(\tau)} \right)$
**end for**
**return** $\theta_{(\tau)}$

---

## B   PARALLEL VECTOR FIELDS AND PARALLEL TRANSFORM IN RIEMANNIAN MANIFOLDS

We give additional explanations of the reason why we cannot define a parallel vector field on a non-Euclidean manifold. Specifically we describe the relationship between parallel vector fields and parallel transform. We can define a parallel transform along a geodesic. This parallel transform maps a tangent vector in a tangent space to one in another. At one glance, it seems that we can define a parallel vector field using the parallel transform. However, a parallel transform is not determined only by the origin and destination but depends on the path i.e. the geodesic. Figure 7 shows an example on a sphere, where two ways to map a vector from a tangent space to another are shown and these two give different maps. As this figure shows we cannot obtain a well-defined vector on more than two points.

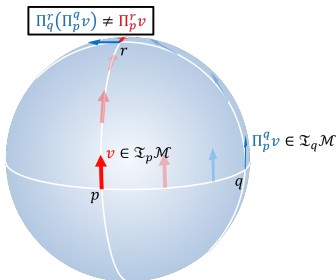

Figure 7: Parallel transforms in a sphere $\mathbb{S}^2$. This figure shows two ways to transform vector $v \in \mathfrak{T}_p\mathbb{S}^2$ to $\mathfrak{T}_r\mathbb{S}^2$. We denote the parallel transform from along segment $pq$ by $\Pi_p^q : \mathfrak{T}_p\mathbb{S}^2 \to \mathfrak{T}_q\mathbb{S}^2$. The red vector on $\mathfrak{T}_r\mathbb{S}^2$ denotes the vector obtained by the direct transform along segment $pr$. The blue vector $\mathfrak{T}_r\mathbb{S}^2$ denotes the vector obtained by the transform via $q$. As this figure shows we cannot obtain a well-defined vector on more than two points.

## C  EXAMPLES OF RIEMANNIAN MANIFOLDS

We introduces some Riemannian manifolds useful in applications, and the formula of the exponential map and logarithmic map in these manifolds. The closed form of exponential map and logarithmic map enables implementation of Riemannian TransE in these manifolds. In the following, we omit symbols $\frac{\partial}{\partial x}$ and d x of the basis in a tangent and cotangent space, respectively, for notation simplicity. Moreover, we give the composition of the exponential map and index raising and that of the index lowering and logarithmic map instead of the exponential map and logarithmic map themselves. This is because we use a cotangent vector rather than a tangent vector in a practical implementation and map from/to cotangent space is more useful (Recall that $\frac{\partial}{\partial \boldsymbol{\theta}^\top}\tilde{\mathcal{L}}$ is not the coordinate of a tangent but the coordinate of a cotangent vector).

### C.1  EUCLIDEAN SPACE

In a $D$-dimensional Euclidean Space, the exponential map (with the index raising) $\mathrm{Exp}_{\boldsymbol{p}} \circ \sharp : \mathfrak{T}_{\boldsymbol{p}}^*\mathbb{R}^D \to \mathbb{R}^D$ is given by

$$\left(\mathrm{Exp}_{\boldsymbol{p}} \circ \sharp\right)(\boldsymbol{\delta}) = \boldsymbol{p} + \boldsymbol{\delta}. \tag{10}$$

Apparently, the logarithmic map (with the index lowering) $\flat \circ \mathrm{Log}_{\boldsymbol{p}} : \mathbb{R}^D \to \mathfrak{T}_{\boldsymbol{p}}^*\mathbb{R}^D$ is given by

$$\left(\flat \circ \mathrm{Log}_{\boldsymbol{p}}\right)(\boldsymbol{q}) = \boldsymbol{q} - \boldsymbol{p}. \tag{11}$$

### C.2  SPHERE

A $D$-dimensional (unit) sphere is given by point set $\mathbb{S}^D := \left\{\boldsymbol{p} \in \mathbb{R}^{(D+1)} \,\middle|\, \boldsymbol{p}^\top\boldsymbol{p} = 1\right\}$, and the cotangent space $\mathfrak{T}_{\boldsymbol{p}}^*\mathbb{S}^D$ on $\boldsymbol{p} \in \mathbb{S}^D$ is identified with $\left\{\boldsymbol{\delta} \in \mathbb{R}^{(D+1)} \,\middle|\, \boldsymbol{p}^\top\boldsymbol{\delta} = 0\right\}$. The distance $\Delta(\boldsymbol{p}, \boldsymbol{q})$ between two points $\boldsymbol{p} \in \mathbb{S}^D$ and $\boldsymbol{q} \in \mathbb{S}^D$ is given as follows:

$$\Delta(\boldsymbol{p}, \boldsymbol{q}) = \arccos\left(\boldsymbol{p}^\top\boldsymbol{q}\right), \tag{12}$$

where $\arccos : [-1, 1] \to [0, \pi]$ denote arc-cosine function. The exponential map (with the index raising) $\mathrm{Exp}_{\boldsymbol{p}} \circ \sharp : \mathfrak{T}_{\boldsymbol{p}}^*\mathbb{S}^D \to \mathbb{S}^D$ is given by

$$\left(\mathrm{Exp}_{\boldsymbol{p}} \circ \sharp\right)(\boldsymbol{\delta}) = \cos\left(\sqrt{\boldsymbol{\delta}^\top\boldsymbol{\delta}}\right)\boldsymbol{p} + \mathrm{sinc}\left(\sqrt{\boldsymbol{\delta}^\top\boldsymbol{\delta}}\right)\boldsymbol{\delta}, \tag{13}$$

where $\mathrm{sinc}$ denotes the cardinal sine function defined as follows:

$$\mathrm{sinc}\,x = \begin{cases} \frac{\sin x}{x} & \text{if } x \neq 0 \\ 1 & \text{if } x = 0. \end{cases} \tag{14}$$

The logarithmic map (with the index lowering) $\flat \circ \mathrm{Log}_{\boldsymbol{p}} : \mathbb{S}^D \to \mathfrak{T}^*_{\boldsymbol{p}} \mathbb{S}^D$ is given by

$$\left(\flat \circ \mathrm{Log}_{\boldsymbol{p}}\right)(\boldsymbol{q}) = \frac{\arccos\left(\boldsymbol{p}^\top \boldsymbol{q}\right)}{\sqrt{1 - \left(\boldsymbol{p}^\top \boldsymbol{q}\right)^2}} \left(\boldsymbol{q} - \left(\boldsymbol{p}^\top \boldsymbol{q}\right)\boldsymbol{p}\right). \tag{15}$$

Note that in optimization, we need the projection of the differential $\tilde{\boldsymbol{\delta}} = \frac{\partial}{\partial \boldsymbol{\theta}} \mathcal{L}(\boldsymbol{\theta})|_{\boldsymbol{\theta}=\boldsymbol{p}}$ of the loss function $\mathcal{L}$ to cotangent vector $\boldsymbol{\delta}$ given by:

$$\boldsymbol{\delta} = \tilde{\boldsymbol{\delta}} - \left(\boldsymbol{p}^\top \tilde{\boldsymbol{\delta}}\right)\boldsymbol{p}. \tag{16}$$

## C.3  HYPERBOLIC SPACE

In this subsection, we introduces models of a hyperbolic space, which are mathematically equivalent to each other, but have practically different aspects. There are many models of a hyperbolic space. We introduce two of them: the hyperboloid model and Poincaré disk model.

### C.3.1  HYPERBOLOID MODEL

Some formulae here are also given and used in Nickel & Kiela (2018). Let $\boldsymbol{G}_{\mathrm{M}}$ denote diagonal matrix

$$\boldsymbol{G}_{\mathrm{M}} := \begin{bmatrix} -1 & & & \\ & 1 & & \\ & & \ddots & \\ & & & 1 \end{bmatrix} \in \mathbb{R}^{(D+1)\times(D+1)} \tag{17}$$

Let $\langle \cdot, \cdot \rangle_{\mathrm{M}} : \mathbb{R}^{(D+1)} \times \mathbb{R}^{(D+1)} \to \mathbb{R}$ denote the Minkowski inner product defined by

$$\langle \boldsymbol{p}, \boldsymbol{q} \rangle_{\mathrm{M}} := \boldsymbol{p}^\top \boldsymbol{G}_{\mathrm{M}} \boldsymbol{q} = -p^0 q^0 + \sum_{d=1}^{D} p^d q^d, \text{ for } \boldsymbol{p} = \begin{bmatrix} p^0 \\ p^1 \\ \vdots \\ p^D \end{bmatrix}, \boldsymbol{q} = \begin{bmatrix} q^0 \\ q^1 \\ \vdots \\ q^D \end{bmatrix}. \tag{18}$$

In the hyperboloid model, a (canonical) hyperbolic space is given by point set $\mathbb{H}^D := \left\{\boldsymbol{p} \in \mathbb{R}^{D+1} \mid \langle \boldsymbol{p}, \boldsymbol{p} \rangle_{\mathrm{M}} = -1, p^0 > 0\right\}$. The tangent space $\mathfrak{T}_{\boldsymbol{p}} \mathbb{H}^D$ on $\boldsymbol{p} \in \mathbb{H}^D$ is identified with $\left\{\boldsymbol{\delta} \in \mathbb{R}^{D+1} \mid \langle \boldsymbol{p}, \boldsymbol{\delta} \rangle = 0\right\}$, and the metric $\mathfrak{g}_{\boldsymbol{p}} : \mathfrak{T}_{\boldsymbol{p}} \mathbb{H}^D \times \mathfrak{T}_{\boldsymbol{p}} \mathbb{H}^D \to \mathbb{R}$ in the tangent space is given by $\mathfrak{g}_{\boldsymbol{p}}(\boldsymbol{u}, \boldsymbol{v}) = \langle \boldsymbol{u}, \boldsymbol{v} \rangle_{\mathrm{M}}$. Hence, the cotangent space $\mathfrak{T}^*_{\boldsymbol{p}} \mathbb{H}^D$ on $\boldsymbol{p} \in \mathbb{H}^D$ is identified with $\left\{\boldsymbol{\delta} \in \mathbb{R}^{D+1} \mid \boldsymbol{p}^\top \boldsymbol{\delta} = 0\right\}$, and the metric $\mathfrak{g}^*_{\boldsymbol{p}} : \mathfrak{T}^*_{\boldsymbol{p}} \mathbb{H}^D \times \mathfrak{T}^*_{\boldsymbol{p}} \mathbb{H}^D \to \mathbb{R}$ in the cotangent space is given by $\mathfrak{g}^*_{\boldsymbol{p}}(\boldsymbol{\gamma}, \boldsymbol{\delta}) = \langle \boldsymbol{\gamma}, \boldsymbol{\delta} \rangle_{\mathrm{M}}$. Note that $\boldsymbol{\delta} \in \mathfrak{T}^*_{\boldsymbol{p}} \mathbb{H}^D$ is identified with $\boldsymbol{\delta}^\sharp = \boldsymbol{G}_{\mathrm{M}}^{-1} \boldsymbol{\delta} \in \mathfrak{T}_{\boldsymbol{p}} \mathbb{H}^D$. The distance $\Delta(\boldsymbol{p}, \boldsymbol{q})$ between two points $\boldsymbol{p} \in \mathbb{H}^D$ and $\boldsymbol{q} \in \mathbb{H}^D$ is given as follows:

$$\Delta(\boldsymbol{p}, \boldsymbol{q}) = \mathrm{arcosh}\left(-\langle \boldsymbol{p}, \boldsymbol{q} \rangle_{\mathrm{M}}\right), \tag{19}$$

where, $\mathrm{arcosh} : [1, \infty) \to [0, \infty)$ denotes the area hyperbolic cosine function, i.e. the inverse fucntion of the hyperbolic cosine function. The exponential map (with the index raising) $\mathrm{Exp}_{\boldsymbol{p}} \circ \sharp : \mathfrak{T}^*_{\boldsymbol{p}} \mathbb{H}^D \to \mathbb{H}^D$ is given by

$$\left(\mathrm{Exp}_{\boldsymbol{p}} \circ \sharp\right)(\boldsymbol{\delta}) = \cosh\left(\sqrt{\langle \boldsymbol{\delta}, \boldsymbol{\delta} \rangle_{\mathrm{M}}}\right)\boldsymbol{p} + \mathrm{sinhc}\left(\sqrt{\langle \boldsymbol{\delta}, \boldsymbol{\delta} \rangle_{\mathrm{M}}}\right)\boldsymbol{G}_{\mathrm{M}}^{-1}\boldsymbol{\delta}. \tag{20}$$

where $\mathrm{sinhc}$ denotes the hyperbolic sine cardinal function defined as follows:

$$\mathrm{sinhc}\, x = \begin{cases} \frac{\sinh x}{x} & \text{if } x \neq 0 \\ 1 & \text{if } x = 0. \end{cases} \tag{21}$$

The logarithmic map (with the index lowering) $\flat \circ \mathrm{Log}_{\boldsymbol{p}} : \mathbb{H}^D \to \mathfrak{T}^*_{\boldsymbol{p}} \mathbb{H}^D$ is given by

$$\left(\flat \circ \mathrm{Log}_{\boldsymbol{p}}\right)(\boldsymbol{q}) = \boldsymbol{G}_{\mathrm{M}} \frac{\mathrm{arcosh}\left(-\langle \boldsymbol{p}, \boldsymbol{q} \rangle_{\mathrm{M}}\right)}{\sqrt{\langle \boldsymbol{p}, \boldsymbol{q} \rangle_{\mathrm{M}}^2 - 1}} \left(\boldsymbol{q} + \langle \boldsymbol{p}, \boldsymbol{q} \rangle_{\mathrm{M}} \boldsymbol{p}\right). \tag{22}$$

Note that in optimization, we need the projection of the differential $\tilde{\boldsymbol{\delta}} = \frac{\partial}{\partial \boldsymbol{\theta}} \mathcal{L}(\boldsymbol{\theta})|_{\boldsymbol{\theta}=\boldsymbol{p}}$ of the loss function $\mathcal{L}$ to cotangent vector $\boldsymbol{\delta}$ given by:

$$\boldsymbol{\delta} = \tilde{\boldsymbol{\delta}} + \boldsymbol{G}_{\mathrm{M}} \left\langle \boldsymbol{p}, \boldsymbol{G}_{\mathrm{M}}^{-1} \tilde{\boldsymbol{\delta}} \right\rangle_{\mathrm{M}} \boldsymbol{p}. \tag{23}$$

### C.3.2 POINCARÉ DISK MODEL

In the Poincaré disk model, the $D$-dimensional hyperbolic space is given by the unit open hyper-ball $\mathbb{D}^D := \left\{ \boldsymbol{p} \in \mathbb{R}^D \mid \boldsymbol{p}^\top \boldsymbol{p} < 1 \right\}$. The Poincaré disk model and the hyperboloid model are derived from each other by the following map:

$$\mathbb{H}^D \ni \boldsymbol{p} = \begin{bmatrix} p^0 \\ p^1 \\ \vdots \\ p^D \end{bmatrix} \mapsto \frac{1}{1+p^0} \begin{bmatrix} p^1 \\ p^2 \\ \vdots \\ p^D \end{bmatrix} \in \mathbb{D}^D$$

$$\mathbb{D}^D \ni \boldsymbol{p} = \begin{bmatrix} p^1 \\ p^2 \\ \vdots \\ p^D \end{bmatrix} \mapsto \frac{1}{\mu_{\boldsymbol{p}}} \begin{bmatrix} 1-\mu_{\boldsymbol{p}} \\ p^1 \\ \vdots \\ p^D \end{bmatrix} \in \mathbb{H}^D, \tag{24}$$

where $\mu_{\boldsymbol{p}} := \frac{1-\boldsymbol{p}^\top \boldsymbol{p}}{2}$.

The metric is given by $\mathfrak{g}_{\boldsymbol{p}}(\boldsymbol{u}, \boldsymbol{v}) = \left( \frac{2}{1-\boldsymbol{p}^\top \boldsymbol{p}} \right)^2 \boldsymbol{u}^\top \boldsymbol{v}$. The distance $\Delta(\boldsymbol{p}, \boldsymbol{q})$ between two points $\boldsymbol{p} \in \mathbb{H}^D$ and $\boldsymbol{q} \in \mathbb{H}^D$ is given as follows:

$$\Delta(\boldsymbol{p}, \boldsymbol{q}) = \operatorname{arcosh}(1 + M), \tag{25}$$

where

$$M := \frac{2(\boldsymbol{q}-\boldsymbol{p})^\top (\boldsymbol{q}-\boldsymbol{p})}{(1-\boldsymbol{p}^\top \boldsymbol{p})(1-\boldsymbol{q}^\top \boldsymbol{q})}. \tag{26}$$

The exponential map (with index raising) $\operatorname{Exp}_{\boldsymbol{p}} \circ \sharp : \mathfrak{T}_{\boldsymbol{p}}^* \mathbb{D}^D \to \mathbb{D}^D$ is given by

$$\left( \operatorname{Exp}_{\boldsymbol{p}} \circ \sharp \right)(\boldsymbol{\delta}) = \left[ 1 - \frac{\mu_{\boldsymbol{p}} \left( \operatorname{sech} \sqrt{\mu_{\boldsymbol{p}}^2 \boldsymbol{\delta}^\top \boldsymbol{\delta}} - 1 \right)}{\beta} \right] \boldsymbol{p} + \frac{\mu_{\boldsymbol{p}}^2 \operatorname{tanhc} \sqrt{\mu_{\boldsymbol{p}}^2 \boldsymbol{\delta}^\top \boldsymbol{\delta}}}{\beta} \boldsymbol{\delta}, \tag{27}$$

where

$$\beta := \mu_{\boldsymbol{p}} \operatorname{sech} \sqrt{\mu_{\boldsymbol{p}}^2 \boldsymbol{\delta}^\top \boldsymbol{\delta}} + \left( 1 - \mu_{\boldsymbol{p}} \right) + \mu_{\boldsymbol{p}} \left( \boldsymbol{p}^\top \boldsymbol{\delta} \right) \operatorname{tanhc} \sqrt{\mu_{\boldsymbol{p}}^2 \boldsymbol{\delta}^\top \boldsymbol{\delta}}. \tag{28}$$

The logarithmic map (with index lowering) $\flat \circ \operatorname{Log}_{\boldsymbol{p}} : \mathbb{D}^D \to \mathfrak{T}_{\boldsymbol{p}}^* \mathbb{D}^D$ is given by

$$\left( \flat \circ \operatorname{Log}_{\boldsymbol{p}} \right)(\boldsymbol{q}) = \left[ \frac{\operatorname{arcosh}(1+M) \mu_{\boldsymbol{p}} \left( \mu_{\boldsymbol{q}}(\boldsymbol{q}-\boldsymbol{p}) - M\boldsymbol{p} \right)}{\sqrt{M} \sqrt{M+2}} \right]^\top \mathrm{d}\boldsymbol{x}. \tag{29}$$

These formulae can be obtained by the coordinate transformation and can be interpreted as a modification of existing formulae such as ones in Ganea et al. (2018a). In addition, these formulae are useful in an automatic differentiation system, because $\operatorname{sech} \sqrt{x}$, $\operatorname{tanhc} \sqrt{x}$, and $\frac{\operatorname{arcosh}(1+x)}{\sqrt{x}}$, and their derivatives do not diverge when $x \to 0$.

## D DETAILS OF EXPERIMENTS

### D.1 EVALUATION TASKS

We evaluated the performance of our method in the link prediction Bordes et al. (2013) and task and the triple classification task Socher et al. (2013) on real knowledge base data sets.

### D.1.1   LINK PREDICTION TASK

In the link prediction task, we predict the head or the tail entity given the relation type and the other entity. We evaluate the ranking of each correct test triple $(h, r, t)$ in the corrupted triples. We corrupt each triple as follows. In our setting, either its head or tail is replaced by one of the possible head or entity, respectively. In addition, we applied "filtered" setting proposed by Bordes et al. (2013), where the correct triples, that is, the triples $\mathcal{T}$ in the original multi-relational graph are excluded. Thus, the corrupted triples are given by $\left\{(h', r, t) \mid h' \in \mathcal{V}^{\mathrm{h}} \wedge (h', r, t) \notin \mathcal{T}\right\}$ (head corruption) or $\left\{(h, r, t') \mid t' \in \mathcal{V}^{\mathrm{t}} \wedge (h, r, t') \notin \mathcal{T}\right\}$ (tail corruption). where $\mathcal{V}_r^{\mathrm{h}}$ and $\mathcal{V}_r^{\mathrm{t}}$ denote the possible heads and tails in relation $r$, given as follows:

$$
\begin{aligned}
\mathcal{V}_r^{\mathrm{h}} &:= \{h \in \mathcal{V} \mid \exists t : (h, r, t) \in \mathcal{T}\}, \\
\mathcal{V}_r^{\mathrm{t}} &:= \{t \in \mathcal{V} \mid \exists h : (h, r, t) \in \mathcal{T}\}.
\end{aligned}
\tag{30}
$$

As evaluation metrics, we use the following:

**Mean rank (MR)**   the mean rank of the correct test triples. The value of this metric is always equal to or greater than 1, and the lower, the better.

**Hits @ $n$ (@$n$)**   the propotion of correct triples ranked in the top $n$ predictions ($n = 1, 3, 10$). The value ranges from 0 to 1, and the higher, the better.

**Mean reciprocal rank (MRR)**   the mean of the reciprocal rank of the correct test triples. The value ranges from 0 to 1, and the higher, the better.

### D.1.2   TRIPLE CLASSIFICATION TASK

In triple classification tasks, we predict whether a triple in the test data is correct or not. The classification is simply based on the score function i.e. we label a triple positive when $f\left(p_h, p_t; (\ell_r, p_r)\right) > \theta_r$, and the other way around. Here, $\theta_r \in \mathbb{R}_{\geq 0}$ denotes the threshold for each relation $r$, which is determined by the accuracy in the validation set.

### D.2   DATASETS

In link prediction tasks, we used WN18 and FB15k Bordes et al. (2013) datasets, and WN11 and FB13 datasets Socher et al. (2013). In triple classification tasks, we used WN11 and FB13 datasets, as well as FB15k. Note that WN18 and FB15k are originally used for link prediction tasks, whereas WN11 and FB13 are originally used for triple classification tasks. Also note that WN18 cannot be used for the triple classification task because WN18 does not have test negative data. Table 3 shows the number of the entities, relations, and triples in each dataset.

Table 3: Statistics of the experimental datasets

| Dataset | $\lvert\mathcal{V}\rvert$ | $\lvert\mathcal{R}\rvert$ | # triples | | |
| --- | --- | --- | --- | --- | --- |
| | | | train | valid | test |
| **WN18** | 40943 | 18 | 141442 | 5000 | 5000 |
| **FB15k** | 14951 | 1345 | 483142 | 50000 | 59071 |
| **WN11** | 38696 | 11 | 112581 | 2609 | 10544 |
| **FB13** | 70543 | 13 | 316232 | 5908 | 23733 |

**Manifolds in Riemannian TransE**   To evaluate the dependency of performance of Riemannian TransE, we compared Riemannian TransE using the following five kinds of manifolds: Euclidean space $\mathbb{R}^D$ (Euclidean TransE), hyperbolic spaces $\mathbb{H}^D$ (HyperbolicTransE), spheres $\mathbb{S}^D$ (Spherical-TransE), the direct product $\mathbb{H}^4 \times \mathbb{H}^4 \times \cdots \times \mathbb{H}^4$ of hyperbolic spaces (PHyperbolicTransE), and the direct product $\mathbb{S}^4 \times \mathbb{S}^4 \times \cdots \times \mathbb{S}^4$ of spheres (PSphericalTransE).

### D.3   BASELINES AND IMPLEMENTATION

We compared our method with baselines. As baselines, we used RESCAL Nickel et al. (2011), TransE Bordes et al. (2013), TransH Wang et al. (2014), TransR Lin et al. (2015), TransD Ji

et al. (2016), DISTMULT Yang et al. (2015), HolE Nickel et al. (2016) and ComplEx Trouillon et al. (2016). We used implementations of the baselines in OpenKE `http://openke.thunlp.org/static/index.html`, a Python library of knowledge base embedding based on Tensorflow Abadi et al. (2015), and moreover, we implemented some lacked constraints (for example, in TransR, TransD) and regularizers (for example, in DistMult, Analogy) in OpenKE. We also found that omitting the constraint of the entity planets onto sphere in TransE gives much better results in our setting, and this is why we also show the result without the constraint (UnconstraintTransE). We also implemented Riemannian TransEs as derivations of the base class of OpenKE.

We set the dimensionality of the entity manifold as $D = 8, 16, 32, 64, 128$. Although we also have to determine the dimensionality of the projected space in TransR and TransD, we let them be equal to $D$. Due to limitation of the computational costs, we fixed the batch size in baselines and Riemannian TransEs such that that the training data are split to 100 batches. We also fixed the number of epochs to 1000. Note that in the first 100 epochs in Riemannian TransEs, we fixed the launchers. Also note that we applied norm clipping such that the norm of a stochastic gradient in the tangent space is smaller than 1. We did not use "bern" setting introduced in Wang et al. (2014), where the ratio between head and tail corruption is not fixed to one to one; in other words, we replaced head and tail with equal probability.

Other than the dimensionality and batch sizes, we used hyperparameters such as learning rate $\eta$ and margin paremeter $\delta$ of baselines used in each paper. Note that some methods only reports link prediction tasks, and reports hyperparameters for WN18 and FB15k and do not reports ones for WN11 and FB13. Some methods do not mention settings of hyperparameters, and in these cases, we used the default parameters in OpenKE. In these cases, we used hyperparameters of WN18 and FB15k also for WN11 and FB13, respectively. Note that the parameters of TorusE is supposed to be used with very high dimensionality, and the hyperparameters are designed for high dimensionality settings. In Riemannian TransEs, we simply followed the hyperparameters in TransE.

We used the Xavier initializer Glorot & Bengio (2010) as an initializer. When we have to use the points on a sphere (in the original TransE and Spherical TransEs), we projected the points generated by the initialization onto the sphere. We found that choice of an initializer has significant effect on embedding performance, and the Xavier initializer achieves very good performance.

We selected optimizers in baselines following each paper. Note that while using ADADELTA (Zeiler, 2012) is also proposed in TransD, we used SGD in TransD. In Riemannian TransEs, we used we simply followed the hyperparameters in TransE. Table 4 shows the hyperparameters and optimization method for each method.

Table 4: Hyperparameters and optimizers: SGD denotes the stochastic gradient descent method (in a Euclidean space). SRGD denotes the stochastic Riemannian gradient descent method Zhang & Sra (2016) with gradient clipping. Adagrad is proposed by Duchi et al. (2011).

| Method | Optimizer | Learning rate $\eta$ | | | | Margin $\delta$ | | | |
|---|---|---|---|---|---|---|---|---|---|
| | | WN18 | FB15k | WN11 | FB13 | WN18 | FB15k | WN11 | FB13 |
| RiemannianTransEs | SRGD | 0.01 | 0.01 | 0.01 | 0.01 | 2.0 | 1.0 | 2.0 | 1.0 |
| TransE | SGD | 0.01 | 0.01 | 0.01 | 0.01 | 2.0 | 1.0 | 2.0 | 1.0 |
| TransH | SGD | 0.01 | 0.005 | 0.001 | 0.005 | 1.0 | 0.5 | 2.0 | 0.25 |
| TransR | SGD | 0.01 | 0.005 | 0.001 | 0.005 | 4.0 | 1.0 | 4.0 | 2.0 |
| TransD | SGD | 0.01 | 0.01 | 0.01 | 0.01 | 1.0 | 1.0 | 1.0 | 1.0 |
| TorusE | SGD | 0.0005 | 0.001 | 0.0005 | 0.001 | 2000.0 | 500.0 | 2000.0 | 500.0 |
| RESCAL | Adagrad | 0.1 | 0.1 | 0.1 | 0.1 | 1.0 | 1.0 | 1.0 | 1.0 |
| DistMult | Adagrad | 0.1 | 0.1 | 0.1 | 0.1 | 1.0 | 1.0 | 1.0 | 1.0 |
| ComplEx | Adagrad | 0.5 | 0.5 | 0.5 | 0.5 | 1.0 | 1.0 | 1.0 | 1.0 |
| HolE | Adagrad | 0.1 | 0.1 | 0.1 | 0.1 | 1.0 | 1.0 | 1.0 | 1.0 |
| Analogy | Adagrad | 0.1 | 0.1 | 0.1 | 0.1 | 1.0 | 1.0 | 1.0 | 1.0 |

## D.4 RESULTS

Table 5 shows the results of triple classification tasks in FB15k. In FB15k, the baselines such as TransH, ComplEx and Analogy attained good accuracies and the Riemannian TransEs did not out-

perform the baselines. Table 6, Table 7, and Table 8 shows hit@10, mean rank, and mean reciprocal rank score of link prediction tasks, respectively. As in triple classification tasks, the sphere-based Riemannian TransEs achieved good accuracy in WN11, whereas the hyperbolic-space-based Riemannian TransEs was more accurate than other methods in FB13. The Riemannian TransEs did not outperform the baselines in WN18 and FB15k. This tendency is apparent in MR score. The distance-based methods such as TransE, TransH and Riemannian TransEs tend to attain good scores in MR and the inner-product-based methods such as DistMult, ComplEx and Analogy tend to attain good scores in MRR and hit@10.

## D.5 ADDITIONAL DISCUSSION

Why do these baselines attain good results in WN18 and FB15k but bad results in WN11 and FB13? One reason may simply be that WN18 and FB15k datasets have good compatibility with zero curvature spaces i.e. Euclidean space. This is supported by the results of Euclidean TransE. A possible second reason is the redundancy of FB15k. Whereas some "easy" relations are excluded from FB15k Bordes et al. (2013), it still contain many reversible triples, as noted by Toutanova & Chen (2015). By contrast, these are removed in WN11 and FB13. Recall that projection-based methods such as TransH, TransR and TransD, and inner-product-based methods such as ComplEx and DISTMULT can exploit a linear subspace. When a dataset has apparent clusters inside which one relation is easily recovered from the others, we can allocate each cluster to a subspace and separate subspaces from one another. This separation is easily realized by setting some elements in the launchers to zero in these methods. Indeed, the TransE without the sphere constraint attains good accuracies in WN11 and FB13.

Differences between criteria are also interesting phenomena. Note that MRR and hit@10 is generous for heavy mistakes. It is possible that inner-product-based methods earn good scores in trivial relations, but further intensive investigation is needed.

Table 5: Triple classification performance. **Bold**: Top 1, *Italic*: Top 3.

| Dataset | **FB15K** | | | | |
|---|---|---|---|---|---|
| Dim. | 8 | 16 | 32 | 64 | 128 |
| **Hyperbolic TransE** | 75.46 | 76.86 | 77.34 | 77.73 | 77.87 |
| **PHyperbolic TransE** | 76.78 | 81.40 | 85.89 | 89.33 | *91.13* |
| **Spherical TransE** | 68.43 | 68.36 | 70.12 | 68.51 | 70.28 |
| **PSpherical TransE** | 74.38 | 79.73 | 84.31 | 88.39 | *90.31* |
| **Euclidean TransE** | **79.46** | *83.31* | 87.22 | *90.11* | **91.52** |
| TransE | 74.02 | 78.72 | 81.05 | 80.50 | 76.67 |
| TransE (unconstraint) | 78.05 | 81.72 | 84.42 | 85.45 | 84.74 |
| TorusE | 56.17 | 56.09 | 56.15 | 56.10 | 56.22 |
| TransH | *78.10* | 82.74 | 85.83 | 87.33 | 87.82 |
| TransR | 69.85 | 75.09 | 77.65 | 78.01 | 75.53 |
| TransD | 56.44 | 60.11 | 63.12 | 66.17 | 71.87 |
| RESCAL | 77.66 | 81.36 | 84.08 | 83.71 | 81.10 |
| DistMult | 77.13 | 82.89 | *88.19* | 89.64 | 89.90 |
| ComplEx | *78.72* | **85.66** | **89.22** | **90.37** | 89.75 |
| HolE | 68.87 | 73.61 | 78.37 | 83.80 | 86.12 |
| Analogy | 76.41 | *83.87* | *88.23* | *89.75* | 90.13 |

Table 6: hit@10 in link prediction task. **Bold**: Top 1, *Italic*: Top 3.

| WN18 (hit@10) / dim | 8 | 16 | 32 | 64 | 128 |
|---|---|---|---|---|---|
| **Hyperbolic TransE** | 21.73 | 31.88 | 38.09 | 41.55 | 40.93 |
| **PHyperbolic TransE** | 29.18 | *75.03* | 85.15 | 86.52 | 87.56 |
| **Spherical TransE** | 32.16 | 31.38 | 33.63 | 34.80 | 36.18 |
| **PSpherical TransE** | 30.13 | 55.56 | 85.82 | 92.83 | 93.22 |
| **Euclidean TransE** | *38.37* | *75.05* | 84.84 | 86.50 | 87.17 |
| TransE | 12.35 | 17.80 | 19.56 | 17.00 | 11.19 |
| TransE (unconstraint) | *37.36* | 66.23 | 86.34 | 88.82 | 86.94 |
| TorusE | 19.45 | 30.04 | 31.62 | 36.52 | 36.85 |
| TransH | **42.11** | **76.36** | 88.34 | 92.41 | 90.72 |
| TransR | 00.89 | 01.32 | 04.95 | 18.57 | 43.19 |
| TransD | 03.23 | 10.64 | 23.65 | 66.14 | 92.14 |
| RESCAL | 14.91 | 39.31 | 74.80 | 81.98 | 77.26 |
| DistMult | 15.69 | 63.18 | *93.95* | *94.09* | *94.08* |
| ComplEx | 19.29 | 73.79 | *93.99* | *94.20* | *93.88* |
| HolE | 10.54 | 08.64 | 15.00 | 28.59 | 81.44 |
| Analogy | 12.77 | 70.67 | **94.20** | **94.22** | **94.33** |

| FB15K (hit@10) / dim | 8 | 16 | 32 | 64 | 128 |
|---|---|---|---|---|---|
| **Hyperbolic TransE** | *44.45* | 48.70 | 50.85 | 51.94 | 52.37 |
| **PHyperbolic TransE** | 43.17 | 51.67 | *61.38* | 71.86 | 79.35 |
| **Spherical TransE** | 34.92 | 34.98 | 37.60 | 38.47 | 39.98 |
| **PSpherical TransE** | 40.17 | 49.38 | 59.45 | 70.48 | 77.88 |
| **Euclidean TransE** | **45.52** | **53.82** | **64.15** | *74.91* | 81.11 |
| TransE | 39.10 | 47.01 | 53.82 | 55.85 | 51.86 |
| TransE (unconstraint) | 44.18 | *51.86* | 60.42 | 67.09 | 69.80 |
| TorusE | 18.85 | 19.64 | 20.29 | 19.71 | 19.51 |
| TransH | 42.99 | *52.06* | 61.21 | 70.48 | 75.29 |
| TransR | 31.49 | 40.40 | 47.44 | 50.41 | 49.34 |
| TransD | 20.79 | 25.35 | 31.13 | 37.71 | 44.01 |
| RESCAL | *44.50* | 51.16 | 56.21 | 58.22 | 53.92 |
| DistMult | 38.58 | 45.69 | 58.86 | 74.09 | *83.37* |
| ComplEx | 39.05 | 49.87 | *62.92* | **79.96** | *81.59* |
| HolE | 35.50 | 39.75 | 45.13 | 55.86 | 63.45 |
| Analogy | 38.16 | 47.01 | 58.85 | *74.50* | **83.49** |

| WN11 (hit@10) / dim | 8 | 16 | 32 | 64 | 128 |
|---|---|---|---|---|---|
| **Hyperbolic TransE** | 08.97 | 13.41 | 15.23 | *16.43* | *15.69* |
| **PHyperbolic TransE** | 08.74 | 14.48 | *17.93* | *19.42* | **20.62** |
| **Spherical TransE** | *11.29* | 11.65 | 12.07 | 11.07 | 13.41 |
| **PSpherical TransE** | *11.86* | *17.61* | 17.69 | 12.18 | 07.98 |
| **Euclidean TransE** | 10.47 | *14.65* | *17.80* | **19.87** | *20.57* |
| TransE | 03.28 | 05.71 | 06.45 | 05.14 | 04.34 |
| TransE (unconstraint) | 10.32 | 09.75 | 09.80 | 09.62 | 08.38 |
| TorusE | 07.24 | 09.28 | 10.49 | 10.85 | 10.05 |
| TransH | **16.80** | **18.25** | 13.17 | 08.80 | 07.38 |
| TransR | 00.77 | 01.17 | 01.72 | 01.10 | 02.07 |
| TransD | 00.83 | 01.42 | 02.60 | 05.31 | 04.81 |
| RESCAL | 03.61 | 02.94 | 03.12 | 03.19 | 03.21 |
| DistMult | 02.32 | 03.62 | 03.60 | 02.85 | 02.46 |
| ComplEx | 02.78 | 04.21 | 03.03 | 02.35 | 01.61 |
| HolE | 04.02 | 03.12 | **30.55** | 01.27 | 01.30 |
| Analogy | 03.92 | 03.04 | 03.22 | 03.12 | 02.50 |

| FB13 (hit@10) / dim | 8 | 16 | 32 | 64 | 128 |
|---|---|---|---|---|---|
| **Hyperbolic TransE** | 31.11 | *35.18* | *37.20* | 38.38 | 39.33 |
| **PHyperbolic TransE** | **34.05** | **37.44** | **39.15** | *39.90* | *40.79* |
| **Spherical TransE** | *25.12* | 28.69 | 29.73 | **42.87** | **45.47** |
| **PSpherical TransE** | 32.60 | 33.81 | 33.00 | 33.29 | 33.55 |
| **Euclidean TransE** | 31.85 | 33.91 | *36.36* | *38.43* | *39.49* |
| TransE | 21.57 | 27.39 | 23.37 | 29.01 | 31.40 |
| TransE (unconstraint) | *32.64* | *34.39* | 32.21 | 31.86 | 32.04 |
| TorusE | 16.66 | 17.22 | 16.63 | 17.49 | 17.63 |
| TransH | 18.28 | 22.50 | 26.82 | 28.94 | 29.84 |
| TransR | 14.29 | 12.80 | 13.21 | 13.65 | 13.78 |
| TransD | 15.31 | 13.31 | 15.39 | 17.51 | 18.23 |
| RESCAL | *33.58* | 32.18 | 28.88 | 26.46 | 23.48 |
| DistMult | 23.54 | 22.36 | 21.71 | 19.84 | 17.88 |
| ComplEx | 26.26 | 27.89 | 28.11 | 27.13 | 23.99 |
| HolE | 27.64 | 30.74 | 30.24 | 26.03 | 18.05 |
| Analogy | 23.27 | 23.29 | 22.58 | 19.46 | 17.77 |

Table 7: MR in link prediction task. **Bold**: Top 1, *Italic*: Top 3.

| WN18 (MR) / dim | 8 | 16 | 32 | 64 | 128 |
|---|---|---|---|---|---|
| **Hyperbolic TransE** | 1899.9 | 1388.5 | 1012.3 | 0807.3 | 0839.4 |
| **PHyperbolic TransE** | 0437.9 | *0174.5* | *0152.8* | *0125.9* | **0104.4** |
| **Spherical TransE** | 0653.6 | 0578.5 | 0527.7 | 0536.7 | 0517.8 |
| **PSpherical TransE** | 0763.6 | 0336.9 | **0113.3** | *0175.8* | 0235.4 |
| **Euclidean TransE** | **0225.4** | **0137.5** | *0134.0* | **0120.5** | *0112.7* |
| TransE | 6493.7 | 5836.3 | 6563.7 | 8268.8 | 7798.4 |
| TransE (unconstraint) | *0258.9* | *0200.9* | 0225.5 | 0227.6 | 0257.4 |
| TorusE | 2949.0 | 2398.0 | 2369.5 | 2212.3 | 2257.4 |
| TransH | *0307.0* | 0258.4 | 0295.2 | 0318.2 | 0317.1 |
| TransR | 3095.2 | 3539.9 | 1803.2 | 0638.4 | 0299.0 |
| TransD | 4785.2 | 4450.0 | 4093.0 | 0878.5 | *0233.1* |
| RESCAL | 0501.7 | 0382.6 | 0325.0 | 0360.4 | 0354.2 |
| DistMult | 0444.9 | 0267.2 | 0277.4 | 0270.2 | 0289.1 |
| ComplEx | 0411.5 | 0259.4 | 0267.5 | 0303.7 | 0355.7 |
| HolE | 3755.3 | 2374.6 | 1116.2 | 0900.3 | 0615.1 |
| Analogy | 0592.5 | 0212.0 | 0269.2 | 0297.6 | 0285.5 |

| FB15K (MR) / dim | 8 | 16 | 32 | 64 | 128 |
|---|---|---|---|---|---|
| **Hyperbolic TransE** | 0150.6 | 0135.1 | 0126.0 | 0121.9 | 0120.0 |
| **PHyperbolic TransE** | 0136.2 | 0087.2 | 0053.5 | 0034.3 | *0027.0* |
| **Spherical TransE** | 0191.9 | 0204.4 | 0185.6 | 0189.8 | 0171.7 |
| **PSpherical TransE** | 0136.1 | 0090.7 | 0056.1 | 0036.2 | *0029.0* |
| **Euclidean TransE** | **0098.7** | *0067.0* | *0041.5* | **0029.8** | **0024.9** |
| TransE | 0136.1 | 0091.8 | 0071.3 | 0069.9 | 0090.5 |
| TransE (unconstraint) | *0103.1* | 0070.1 | 0051.7 | 0045.9 | 0050.1 |
| TorusE | 0397.2 | 0400.6 | 0399.7 | 0395.6 | 0391.9 |
| TransH | *0101.7* | *0064.4* | 0044.8 | 0037.7 | 0038.5 |
| TransR | 0178.7 | 0121.2 | 0086.1 | 0070.9 | 0074.2 |
| TransD | 0355.6 | 0304.6 | 0302.7 | 0302.7 | 0219.6 |
| RESCAL | 0103.2 | 0069.6 | 0052.0 | 0051.4 | 0066.8 |
| DistMult | 0127.8 | 0079.8 | 0043.5 | 0032.7 | 0033.4 |
| ComplEx | 0114.6 | **0062.2** | **0036.9** | *0031.7* | 0036.7 |
| HolE | 0226.0 | 0174.5 | 0127.0 | 0076.4 | 0054.8 |
| Analogy | 0129.6 | 0073.3 | *0043.0* | *0032.5* | 0032.2 |

| WN11 (MR) / dim | 8 | 16 | 32 | 64 | 128 |
|---|---|---|---|---|---|
| **Hyperbolic TransE** | 5248.9 | 4974.3 | 4853.2 | 4771.8 | 4824.7 |
| **PHyperbolic TransE** | 4420.0 | 3603.0 | 3115.9 | *2920.1* | *2638.3* |
| **Spherical TransE** | **1856.2** | **1697.9** | **1689.5** | **1609.3** | **1614.0** |
| **PSpherical TransE** | *2059.3* | *2030.0* | *2827.6* | 4128.2 | 5067.7 |
| **Euclidean TransE** | 3466.7 | 3220.4 | *3103.8* | *2772.1* | *2606.1* |
| TransE | 7615.2 | 7642.1 | 7369.9 | 7852.5 | 7872.7 |
| TransE (unconstraint) | 3538.6 | 3791.9 | 4349.4 | 4618.2 | 5145.5 |
| TorusE | 5389.1 | 5329.2 | 5275.6 | 5273.0 | 5335.9 |
| TransH | *2669.1* | *2985.7* | 3952.4 | 4718.7 | 5380.7 |
| TransR | 7291.8 | 6386.8 | 6040.7 | 5924.2 | 5330.1 |
| TransD | 6883.8 | 6437.3 | 6501.1 | 5077.1 | 5321.8 |
| RESCAL | 5395.7 | 5855.6 | 5983.3 | 5997.4 | 6003.5 |
| DistMult | 5320.1 | 5369.7 | 5790.2 | 6079.6 | 6312.5 |
| ComplEx | 5036.6 | 4916.1 | 5791.5 | 6211.7 | 6576.1 |
| HolE | 6681.5 | 6775.8 | 6342.5 | 6777.8 | 6462.2 |
| Analogy | 4802.2 | 5682.7 | 5882.0 | 6106.2 | 6150.7 |

| FB13 (MR) / dim | 8 | 16 | 32 | 64 | 128 |
|---|---|---|---|---|---|
| **Hyperbolic TransE** | 3970.3 | 3391.0 | 3304.6 | 3214.8 | 3215.6 |
| **PHyperbolic TransE** | 2889.3 | 2148.5 | *1835.7* | *1690.3* | **1588.8** |
| **Spherical TransE** | 4563.6 | 3997.7 | 3844.3 | *1869.3* | *1813.7* |
| **PSpherical TransE** | 2304.7 | **1718.1** | *1797.9* | 2238.4 | 2636.6 |
| **Euclidean TransE** | **2072.1** | *1899.7* | **1792.8** | **1675.4** | *1607.5* |
| TransE | 5993.4 | 6341.6 | 6935.9 | 5406.0 | 5462.7 |
| TransE (unconstraint) | 5143.0 | 5508.6 | 6631.5 | 7918.1 | 7639.0 |
| TorusE | 5972.7 | 6081.8 | 6028.5 | 5709.4 | 5468.1 |
| TransH | 4460.7 | 4230.5 | 4386.7 | 4465.0 | 4633.8 |
| TransR | 4474.6 | 5272.6 | 3913.0 | 2654.0 | 2063.4 |
| TransD | 5844.6 | 6643.1 | 6040.2 | 5893.8 | 5893.2 |
| RESCAL | 2356.9 | 2303.4 | 2467.6 | 2744.5 | 3236.0 |
| DistMult | *2202.8* | 2400.3 | 2332.0 | 2452.5 | 2777.8 |
| ComplEx | *2131.2* | *2041.4* | 2261.2 | 2457.6 | 2923.1 |
| HolE | 4695.4 | 3866.8 | 2966.3 | 3332.8 | 3647.3 |
| Analogy | 2313.2 | 2265.1 | 2271.7 | 2469.5 | 2664.6 |

Table 8: MRR in link prediction task. **Bold**: Top 1, *Italic*: Top 3.

| WN18 (MRR) / dim | 8 | 16 | 32 | 64 | 128 |
|---|---|---|---|---|---|
| **Hyperbolic TransE** | 11.07 | 16.55 | 19.59 | 21.00 | 20.49 |
| **PHyperbolic TransE** | 13.77 | 36.26 | 42.57 | 44.18 | 44.94 |
| **Spherical TransE** | 16.42 | 16.22 | 19.12 | 17.81 | 19.04 |
| **PSpherical TransE** | 15.49 | 29.82 | 49.00 | 56.07 | 55.92 |
| **Euclidean TransE** | *17.87* | 35.64 | 42.01 | 44.15 | 44.65 |
| TransE | 06.82 | 09.70 | 11.05 | 09.35 | 06.45 |
| TransE (unconstraint) | *18.73* | 38.72 | 54.88 | 55.44 | 53.00 |
| TorusE | 11.72 | 21.54 | 23.25 | 28.56 | 29.22 |
| TransH | **22.45** | *39.07* | 49.83 | 57.85 | 58.10 |
| TransR | 00.58 | 00.65 | 02.31 | 08.19 | 18.27 |
| TransD | 01.52 | 05.38 | 13.30 | 39.37 | 55.03 |
| RESCAL | 08.58 | 20.78 | 46.32 | 61.98 | 60.63 |
| DistMult | 08.14 | 34.52 | *76.38* | *83.72* | *83.67* |
| ComplEx | 09.72 | **43.12** | *80.42* | *92.25* | *92.90* |
| HolE | 07.27 | 06.02 | 08.20 | 17.00 | 58.33 |
| Analogy | 06.83 | *41.48* | **81.11** | **93.22** | **93.99** |

| FB15K (MRR) / dim | 8 | 16 | 32 | 64 | 128 |
|---|---|---|---|---|---|
| **Hyperbolic TransE** | *26.60* | 29.92 | 31.79 | 32.74 | 32.94 |
| **PHyperbolic TransE** | 25.81 | 31.14 | *38.23* | *47.36* | 55.43 |
| **Spherical TransE** | 20.05 | 21.17 | 22.82 | 24.09 | 24.83 |
| **PSpherical TransE** | 23.55 | 29.33 | 36.31 | 46.20 | 54.63 |
| **Euclidean TransE** | *26.65* | **32.36** | **40.24** | *49.91* | 57.24 |
| TransE | 22.48 | 27.91 | 32.56 | 33.98 | 30.79 |
| TransE (unconstraint) | 26.53 | *31.76* | 37.28 | 42.26 | 45.18 |
| TorusE | 10.66 | 11.16 | 11.40 | 11.04 | 11.43 |
| TransH | 24.68 | 30.25 | 36.62 | 44.21 | 50.79 |
| TransR | 18.27 | 23.60 | 27.59 | 29.71 | 28.73 |
| TransD | 12.00 | 14.89 | 19.07 | 23.65 | 27.36 |
| RESCAL | **27.84** | *32.19* | 35.06 | 36.19 | 33.14 |
| DistMult | 23.30 | 27.45 | 36.07 | 47.29 | *60.70* |
| ComplEx | 22.93 | 29.24 | *37.77* | **51.84** | *60.47* |
| HolE | 26.14 | 28.44 | 33.31 | 41.68 | 46.73 |
| Analogy | 23.29 | 28.06 | 35.81 | 47.22 | **61.53** |

| WN11 (MRR) / dim | 8 | 16 | 32 | 64 | 128 |
|---|---|---|---|---|---|
| **Hyperbolic TransE** | 04.57 | 06.86 | 07.71 | *08.18* | *07.87* |
| **PHyperbolic TransE** | 04.42 | 07.05 | *08.72* | **09.48** | **09.65** |
| **Spherical TransE** | *05.67* | 06.03 | 06.38 | 05.95 | 07.02 |
| **PSpherical TransE** | *06.16* | *08.86* | *08.56* | 06.37 | 04.37 |
| **Euclidean TransE** | 05.35 | *07.22* | 08.56 | *09.36* | *09.50* |
| TransE | 01.66 | 02.96 | 03.31 | 02.67 | 02.31 |
| TransE (unconstraint) | 05.55 | 05.31 | 05.44 | 05.22 | 04.42 |
| TorusE | 03.66 | 04.84 | 05.98 | 06.30 | 05.62 |
| TransH | **10.22** | **09.88** | 06.61 | 04.51 | 03.82 |
| TransR | 00.49 | 00.73 | 01.03 | 00.64 | 01.05 |
| TransD | 00.55 | 00.85 | 01.50 | 02.82 | 02.52 |
| RESCAL | 01.93 | 01.56 | 01.81 | 01.88 | 01.91 |
| DistMult | 01.16 | 01.92 | 01.93 | 01.58 | 01.36 |
| ComplEx | 01.33 | 02.22 | 01.65 | 01.28 | 00.84 |
| HolE | 03.79 | 02.92 | **30.57** | 00.83 | 00.71 |
| Analogy | 01.91 | 01.58 | 01.77 | 01.67 | 01.32 |

| FB13 (MRR) / dim | 8 | 16 | 32 | 64 | 128 |
|---|---|---|---|---|---|
| **Hyperbolic TransE** | 20.60 | 22.23 | **24.51** | *27.36* | *28.61* |
| **PHyperbolic TransE** | 20.83 | 21.53 | *23.52* | *24.29* | *25.29* |
| **Spherical TransE** | 14.35 | 16.24 | 16.94 | **30.67** | **33.05** |
| **PSpherical TransE** | 18.77 | 18.60 | 18.97 | 19.64 | 21.81 |
| **Euclidean TransE** | *22.84* | *24.08* | 22.54 | 23.48 | 24.05 |
| TransE | 13.71 | 16.62 | 16.48 | 21.66 | 23.07 |
| TransE (unconstraint) | 22.12 | *24.58* | 22.85 | 22.86 | 22.90 |
| TorusE | 11.37 | 09.35 | 12.98 | 11.61 | 09.58 |
| TransH | 12.73 | 14.54 | 17.22 | 19.76 | 21.05 |
| TransR | 12.48 | 07.58 | 08.13 | 09.06 | 09.42 |
| TransD | 12.44 | 10.24 | 08.81 | 10.55 | 11.36 |
| RESCAL | **23.91** | 21.69 | 17.44 | 15.68 | 13.86 |
| DistMult | 13.89 | 14.61 | 14.39 | 13.16 | 11.56 |
| ComplEx | 14.65 | 17.61 | 17.78 | 16.41 | 14.09 |
| HolE | *23.12* | **27.43** | *24.33* | 21.11 | 12.46 |
| Analogy | 14.99 | 15.51 | 14.93 | 13.02 | 11.67 |

