# OpenReview forum: "Riemannian TransE: Multi-relational Graph Embedding in Non-Euclidean Space"
_ICLR.cc/2019/Conference_

### Official Review · AnonReviewer3 · 2018-10-23
**Lack of clarity and limited experimental success**

**Rating:** 5
**Confidence:** 2

**Review:**

This paper presents a generalization of TransE to Riemannian manifolds. While this work falls into the class of interesting recent approaches for using non-Euclidean spaces for knowledge graph embeddings, I found it very hard to digest (e.g. the first paragraph in Section 3.3). Figure 3 and 4 confused me more than helping me to understand the method. Furthermore, current neural link prediction methods are usually evaluated on FB15k and WN18. In fact, often on the harder variants FB15k-237 and WN18RR. For FB15k and WN18, Riemannian TransE seems to underperform compared to baselines -- even for low embedding dimensions, so I have doubts how useful this method will be to the community and believe further experiments on FB15k-237 and WN18RR need to be carried out and the clarity of the paper, particularly the figures, needs to be improved. Lastly, I would be curious about how the different Riemannian TransE variants compare to TransE in terms of speed?

Update: I thank the authors for their response and revision of the paper. To me, results on WN18RR and FB15k-237 are inconclusive w.r.t. to the choice of using Riemannian as opposed to Euclidean space. I therefore still believe this paper needs more work before acceptance.

---

> ### Author Response · Authors · 2018-11-24
> **Author feedback**
>
> We sincerely thank you for the valuable comments and suggestions.
> Question 1：Lack of clarity of the paper
> Response: Thanks for pointing out the ambiguities. We have revised the most parts of the paper for clearer presentation. In particular, we have provided a clearer explanation of definition of exponential map in the caption of Figure 3 which is correspondence to in section 3.1. We have also omitted the redundant notations, e.g., index raising and lowering. To make the structure of whole the paper compactable, we have moved details for optimization, i.e., Algorithm 1, to the Appendix A.  We believe that the revised version has greatly enhanced the clarity of the paper.
>
> Question 2: Underperform results compared to baselines on FB15k and WN18
> Response: Thanks for your comment and suggestion. Yes, our Riemannian TransE does not perform well on FB15k and WN18. This is probably because these two datasets’ structure  do not fit the non-Euclidean manifold but Euclidean space, since all the Euclidean methods including Euclidean TransE work well.
> Nevertheless, according to your suggestion, we have also conducted  experiments on both WN18RR and FB15K237. Our (non-euclidean) methods performs  better than baselines in  WN18RR but slightly worse in  FB15K237. For example, in WN18RR with D=16, spherical TransE achieves the best accuracy with 83.50%, which outperforms the second best approach, Trans H (83.07%), and original TransE (76.32%).  But in  FB15K237 with D=16, original TransE obtains 74.84% which is higher than our hyperbolic TransE (72.24%). However, it is still lower than our Euclidean TransE (76.6%).  We found that the results of these two datasets are in line with that of WN18 and FB15K. This is reasonable since WN18RR and FB15K237 are originated from WN18 and FB15K, respectively. Once again, we could say that with a suitable manifold (Euclidean or non-Euclidean), our methods perform better than baselines.
> Worth to mention that, we did not include the results for these two datasets in the revised version because we aim to ensure consistent parameter settings for all the datasets. For these two datasets, we have to tune parameters (as original papers of some methods do not provide the parameter setting), but we used the parameter settings of each original paper for all the other datasets.
>
> Question 3 : Speed comparison between Riemannian TransE  and TransE
> Response:   The comparison of time complexity relies on two aspects: the sample size and dimensionality.  In terms of the dimensionality,  the Riemannian TransE has linear complexity w.r.t to D-dimensional by using  any of hyperbolic spaces, spheres or  Euclidean spaces, as well as direct products among them.  Hence, it has the same complexity as that of TransE.  In terms of the sample size, the both methods also have the same complexity, as they employ the same loss function (equation (8)) and the same stochastic gradient descent algorithm for optimization.  Indeed, worth to mention that both of the methods have the same algorithm as long as using the same other algorithms, e.g., (Riemannian) SVRG and (Riemannian) Adagrad. To sum up, we can conclude that the Riemannian TransE has the same complexity as TransE.

---

### Official Review · AnonReviewer2 · 2018-11-02
**Very interesting approach, but underwhelming results (despite the complexity)**

**Rating:** 5
**Confidence:** 3

**Review:**

In this paper, authors focus on the problem of efficiently embedding Knowledge Graphs in low-dimensional embedding spaces, a task where models are commonly evaluated via downstream link prediction and triple classification tasks. The proposed model - Riemannian TransE, based on TransE [Bordes et al. 2013] - maps entities to points in a non-Euclidean space, by minimising a loss based on the geodesic distance in such space. This paper is especially interesting, since extends previous approaches - such as Poincare embeddings - to the multi-relational setting. Results look promising on WN11 and FB13, but authors mention results on the more commonly used WN18 and FB15k are less accurate than those obtained by the baselines (without reporting them). It is worth mentioning that WN18 and FB15k were found to be solvable by very simple baselines (e.g. see [1]). Furthermore, authors do not report any finding on the geometry of the learned spaces.

Introduction - Wording is a bit weird sometimes, e.g. what does "evaluating dense matrices or tensors" mean?
Related Work - Likewise, this section was a bit hard to follow. I do not fully get why authors had to use terms like "planet", "launcher", "satellite" etc. for describing mappings between entities and points in a manifold, relations and points in another manifold, and the manifold where the geodesic distances between representations are calculated.
What is the intuition behind this?
Tab. 1 does a great job at summarising existing scoring functions and their space complexity. However, it may be worth noticing that e.g. the inner product used by DistMult is not really a "dissimilarity" between two representations (but rather the opposite). Is the number of parameters the same for Riemannian TransE as for the other methods (including the extra "l" parameters)? If it isn't the comparison may be slightly unfair.

[1] https://arxiv.org/abs/1707.01476

---

> ### Author Response · Authors · 2018-11-24
> **Author feedback**
>
> Thanks very much for your careful reading and constructive feedbacks. We are also greatly encouraged by your insightful comments, e.g. interesting paper and great job at summarizing existing scoring functions.
> Question 1: Results on WN18 and FB15k are less accurate
> Response: Yes, you are right. Our Riemannian TransE does not perform as well as others in  FB15k. However, this is quite reasonable. Since different datasets may have different structure , hence the performance of an approach may largely rely on whether the data structure fits the manifold that it applies on, or otherwise. For example, we can see that Euclidean methods (including our Euclidean TransE) work well in  FB15K (in Table 5), while spherical methods achieve the best in  WN11 (in Table 2) and hyperbolic methods achieve the best in  FB13 (in Table 2). Nevertheless, we admit that a more solid analysis in theory will enhance the paper, which would be our future work.
> Question 2: Introduction-what does "evaluating dense matrices or tensors" mean?
> Response: Thanks for pointing out our wording mistake. To make clearer understanding , we have changed the word “evaluating” to “computing”. In both RESCAL and NTN, each relation corresponds to a dense (non-sparse) matrix or tensor that are  of no less than O(D^2) parameters.
> Question 3: Related Work - confusing for describing mappings using terms like "planet", "launcher", "satellite" etc.
>  Response: Thanks for the comment. For multi-relational data, each entity may contain R relations, where each relation has its own dissimilarity criterion. This just looks like a planet that usually owns R head satellites and R tail satellites, and each paired head-tail satellite differs w.r.t launcher maps. Hence, in this paper, we regard p as "planet", w as "launcher" and s as "satellite", respectively.  Nevertheless, we have revised the description “The idea of …” which is beneath Equation (1), in order to describe the mappings more clearly.
> Question 4:  The inner product used by DistMult is not really a "dissimilarity" between two representations (but rather the opposite).
> Response: Thanks for pointing this out. Yes, you are correct. According to your comment, in the last second paragraph, we have modified “inner product” into “(negative) inner product”. We have also inserted a negative sign for dissimilarity in the last column of the last 5 methods in Table 1.
> Question 5:  Whether the number of parameters is the same for Riemannian TransE as for the other methods
> Response: Thanks for pointing out a good question. According to the first column in Table 1, the number of parameter  for Riemannian TransE is significantly smaller than TransH , TransR and TransD, but is slightly larger than other  approaches. However, since usually |R|<<|V| (For example, in our experiment , |R| is around 10 but |V| is often larger than 100000), hence  the number of parameters for Riemannian TransE and other methods are nearly the same, i.e., D|V|, approximately. In addition, for clearer demonstration  and comparison, we have also added the detailed score function of our Riemannian TransE into the first row of Table 1.

---

### Official Review · AnonReviewer1 · 2018-11-05
**Review - Riemannian TransE**

**Rating:** 5
**Confidence:** 5

**Review:**

The paper proposes a new approach to compute embeddings of multi-relational data such as knowledge graphs. For this purpose, the paper introduces a variant of TransE that operates on Riemannian manifolds, in particular, Euclidean, Spherical, and Hyperbolic space. This approach is motivated by the results of Nickel & Kiela (2017), who showed tha  Hyperbolic space can provide important advantages for embedding graphs with hierarchical structure.

Hyperbolic and Riemannian embeddings are a promising research area that fits well into ICLR. Extending hyperbolic, and more general, Riemannian embeddings to multi-relational data is an important aspect in this context, as it allows to extend such methods to new applications such as Knowledge Graph Completion. Overall, the paper is written well and mostly good to understand. However, I am concerned about multiple aspects in the current version:

- What is the motivation for using this particular form of translation? In Riemannian manifolds, the analogue of vector addition and subtraction is typically taken as the exponential or logarithmic map (as expm and logm in Euclidean space are exactly vector addition and subtraction). For the spherical and hyperbolic manifold both maps have closed form expressions and are differentiable. It is therefore not clear to me what the advantage of the proposed approach is compared to these standard methods. In any case, it would be important to include them in the experimental results.

- It is also not clear to me whether the benefits of hyperbolic embeddings translate into the setting that is proposed here. The advantage of hyperbolic embeddings is that they impose a hierarchical structure in the latent space. The method proposed in this paper uses then a single (hyperbolic) embedding of entities across all relation types. This implies that there should be a single consistent hierarchy that explains all the links in all relations. This seems unlikely and might explain some of the hyperbolic results. A more detailed discussion and motivation would be important here.

- Regarding the experimental results: Why are the results for HolE and ComplEx so different? As [1,2] showed, both models are identical, and for that reason should get close to identical results. The large differences seem inconsistent with these results. Furthermore it seems that the results reported in this paper do not match previously reported results. What is the reason for these discrepancies?

[1] Hayashi, K., and Shimbo, M. "On the equivalence of holographic and complex embeddings for link prediction", 2017.
[2] Trouillon, T, and Nickel, M. "Complex and Holographic Embeddings of Knowledge Graphs: A Comparison", 2017.

---

> ### Author Response · Authors · 2018-11-24
> **Author feedback**
>
> Thank you very much for your time and expertise in reviewing our paper. We address your concerns and questions as follows.
>
> Concern 1: Motivation for in this paper.
> Response:  Thanks for raising a good point.  Worth to mention that, extending TransE into non-Euclidean space cannot be achieved by simply replacing vector addition with exponential map .  Since parallel vector fields exist in Euclidean space, hence all the tangent vectors can be determined once one vector is determined. Unfortunately, no parallel vector fields (See Figure 4 in the revised version) exist in non-Euclidean space,  e.g., sphere and hyperbolic space, and so identifying one vector is hard to guarantee a vector field to be determined . Therefore, the extension requires defining a new vector field. Accordingly, we have to determine the particular form of translation for utilizing the exponential map.
> In addition, we construct the translation according to the two nature  of multi-relational data.  One is the hierarchical  structure of each relation could be seen as “a move towards a top.”  The other is a relation  could be seen as an addition operation (e.g., Rome = Italy + (Paris - France)), and an addition operator could be seen as “a move towards a point at infinity.”  Therefore, we model each relation as one translation in the form of exponential map  along a vector field towards a certain point.
> Thank you for kindly recommending us to compare with a “standard” method. Nevertheless , as  a “standard” method cannot be constructed according to the above explanation, hence  it is infeasible to include its result.
>
> Concern 2: Benefits of using a single (hyperbolic) embedding of entities across all relation types.
> Response:
> Thanks for your comment. Multi-relational data often has multiple hierarchical structures, in which,  each relation can be regarded as “a move towards the top”.  In our model, each relation has its own vector field and attracts all points to its own attraction point. Thus, two relations are close if they have the same attraction point and attraction length, and vice versa (An example is given in Figure 2.). This naturally forms a multi-graph, which is in line with the multi-hierarchical structures. Thus, we argue that it is reasonable to embed all of the entities across all relation types, with utilization  of a single non-Euclidean space.
>
> Concern 3:  1) Different results for HolE and ComplEx
>             2) Discrepancies of our experimental results from previous work
> Response: 1) Thank you for pointing out an important point. Yes, ComplEx and HolE  are "equivalent" in [1], however the "equivalent" cannot always hold.  As we know, the vectors, e.g., [z1,…, zD]^T, in ComplEx are complex D-dimensional,  HolE  in [1] converts a real D-dimensional vectors [x1,…, xD] into a complex D-dimensional [z1,…, zD] with Fourier transform. Hence, if [x1,…, xD] are real, then the corresponding [z1,…, zD] should be symmetric. From this viewpoint, HolE is not "equivalent" to ComplEx but ComplEx  with the symmetricity constraint.  In other words, HolE is a subset of ComplEx.  According to the original paper of HolE [2], HolE also adopts real vectors in our experiment, so the results for HolE and ComplEx are different.
> [1] Hayashi, K., and Shimbo, M. "On the equivalence of holographic and complex embeddings for link prediction", 2017.
> [2] Trouillon, T, and Nickel, M. "Complex and Holographic Embeddings of Knowledge Graphs: A Comparison", 2017.
>
> 2) Though we conducted experiments for each compared method according to their original settings, the discrepancies exist largely due to different reduced dimensionalities between previous work and our experiment .   Previous results are based on a specific dimensionality, e.g., 50 in [1]. However, to validate the effectiveness of the proposed approach on a further reduced the low-dimensional embedding , results in our paper are based on different dimensionalities, e.g., 8 or 16.
>
> [1] Bordes, Antoine, et al. "Translating embeddings for modeling multi-relational data." Advances in neural information processing systems. 2013.

---

### Public Comment · (anonymous) · 2018-09-29
**should report results from original papers**

You should report the experimental results from original papers which are much better than all results (including yours) you reported in Table 2. You can see a part from [1].

[1] An overview of embedding models of entities and relationships for knowledge base completion.

---

> ### Author Response · Authors · 2018-09-29
> **Performance in low dimensionality is our focus**
>
> Thank you for your comment.
> Please kindly note that we focus on relation between accuracy and dimensionality, and specifically on performance in LOW dimensionality.
> This is because good performance in LOW dimensionality is an advantage of using non-Euclidean spaces, as shown in [2].
> On the other hand, most papers (including the paper you referred) only report the result in the best dimensionality after grid search. This is why we had to do experiments by ourselves.
>
> However, as you suggested, noting that some methods can attain better in other dimensionality (though it might be much higher than in our experiments) might be kinder to readers. We'll add a note about that in the revised version.
>
> Thank you.
>
> [2] Nickel, Maximillian, and Douwe Kiela. "Poincaré embeddings for learning hierarchical representations." Advances in neural information processing systems. 2017.

---

> > ### Public Comment · (anonymous) · 2018-09-30
> > **Early models are often evaluated using LOW dimensionality**
> >
> > In the past (even now), early models are often evaluated using LOW dimensionality (might be because of limitations of computer resources). For example, in TransD [1], the embedding size is in {20, 50, 80, 100}, and TransD obtains the accuracies of 86.4% and 89.1% on WN11 and FB13 respectively. Another example is about TransE from [2], TransE gets the accuracy of 85.2% only using the embedding size of 20 on WN11, and gets the accuracy of 87.6% using the embedding size of 100 on FB13.
> >
> > [1] Knowledge Graph Embedding via Dynamic Mapping Matrix. ACL 2015.
> > [2] Neighborhood Mixture Model for Knowledge Base Completion. CoNLL 2016.

---

> > > ### Author Response · Authors · 2018-10-01
> > > **LOW refers to around 10 dimension in our (or other non-Euclidean embeddings) context**
> > >
> > > Please kindly note that in non-Euclidean embedding context such as [3] [4], LOW dimension refers to around 10 dimension. In our context, 200 dimension is TOO HIGH (note that [1] reports the case where m=n=100, i.e. 200 dimensional case). Our results in 64 and 128 dimension are no more than additional information (to observe relation between dimension and accuracy).
> > > Moreover, we found that in their experiments, they used ADADELTA and RMSProp in [1] and [2], respectively, whereas we used SGD for all Trans-X methods for fair comparison (we should distinguish the problem of model and optimizer), which we think is the reason made difference in results (and we suspect that TransD is very sensitive to hyperparameters, datasets and tasks, for TransD gives good results even in our experiments with some settings such as hit@10 in WN18).
> > > However, we admit that we overlooked the results of TransE in 20 dimension in [2].
> > > Although one might think that 20 dimension is still high and our method works well in lower than 20 dimension, it is worth while to note the result in [2]. Thank you for your information.
> > >
> > > [1] Knowledge Graph Embedding via Dynamic Mapping Matrix. ACL 2015.
> > > [2] Neighborhood Mixture Model for Knowledge Base Completion. CoNLL 2016.
> > > [3] Nickel, Maximillian, and Douwe Kiela. "Poincaré embeddings for learning hierarchical representations." Advances in neural information processing systems. 2017.
> > > [4] Ganea, Octavian-Eugen, Gary Bécigneul, and Thomas Hofmann. "Hyperbolic Entailment Cones for Learning Hierarchical Embeddings." ICML 2018.

---

### Meta-Review · Area_Chair1 · 2018-12-17
**Insufficient evidence**

**Confidence:** 4
**Recommendation:** Reject

**Metareview:**

This paper proposes a generalization of the translation-style embedding approaches for link prediction to Riemannian manifolds. The reviewers feel this is an important contribution to the recent work on embedding graphs into non-Euclidean spaces, especially since this work focuses on multi-relational links, thus supporting knowledge graph completion. The results on WN11 and FB13 are also promising.

The reviewers and AC note the following potential weaknesses: (1) the primary concern is the low performance on the benchmarks, especially WN18 and FB15k, and not using the appropriate versions (WN18-RR and FB15k-237), (2) use of hyperbolic embedding for an entity shared across all relations, and (3) lack of discussion/visualization of the learned geometry.

During the discussion phase, the authors clarified reviewer 1's concern regarding the difference in performance between HolE and ComplEx, along with providing a revision that addressed some of the clarity issues raised by reviewer 3. The authors also justified the lower performance due to (1) they are focusing on low-dimensionality setting, and (2) not all datasets will fit the space of the proposed model (like FB15k). However, reviewers 2 and 3 still maintain that the results provide insufficient evidence for the need for Riemannian spaces over Euclidean ones, especially for larger, and more realistic, knowledge graphs.

The reviewers and the AC agree that the paper should not be accepted in the current state.